# The structural basis of the activation and inhibition of DSR2 NADase by phage proteins

Ruiwen Wang[1,5], Qi Xu [2,3,5], Zhuoxi Wu[1], Jialu Li[2,3], Hao Guo[1], Tianzhui Liao[1], Yuan Shi[2,3], Ling Yuan[1], Haishan Gao [2,3], Rong Yang [4] ✉, Zhubing Shi [2,3] ✉ & Faxiang Li [1] ✉

DSR2, a Sir2 domain-containing protein, protects bacteria from phage infection by hydrolyzing NAD⁺. The enzymatic activity of DSR2 is triggered by the SPR phage tail tube protein (TTP), while suppressed by the SPbeta phage-encoded DSAD1 protein, enabling phages to evade the host defense. However, the molecular mechanisms of activation and inhibition of DSR2 remain elusive. Here, we report the cryo-EM structures of apo DSR2, DSR2-TTP-NAD⁺ and DSR2-DSAD1 complexes. DSR2 assembles into a head-to-head tetramer mediated by its Sir2 domain. The C-terminal helical regions of DSR2 constitute four partner-binding cavities with opened and closed conformation. Two TTP molecules bind to two of the four C-terminal cavities, inducing conformational change of Sir2 domain to activate DSR2. Furthermore, DSAD1 competes with the activator for binding to the C-terminal cavity of DSR2, effectively suppressing its enzymatic activity. Our results provide the mechanistic insights into the DSR2-mediated anti-phage defense system and DSAD1-dependent phage immune evasion.

Facing frequent attacks by virus, bacteria and archaea have evolved hundreds of diverse anti-phage defense systems to protect from phage infection, including the well-known restriction-modification (RM), CRISPR-Cas and abortive infection systems[1,2]. The RM and CRISPR-Cas systems are employed to directly degrade invaded phage DNA, providing a robust defense mechanism against phage infections[3]. In contrast, the abortive infection system triggers programmed cell death in infected bacterial cells, sacrificing themselves to prevent phage propagation and ensure the survival of the bacterial population[4]. Bioinformatic analyses and subsequent experimental verification recently discovered plenty of anti-phage systems employing the abortive infection to inhibit phage replication, such as CBASS, toxin-antitoxin, Thoeris, Pycsar and defense-associated sirtuin (DSR) systems[1,5].

To combat the multilayered bacterial immune systems, phages have developed to express anti-defense proteins[6–9]. The most studied phage anti-defense proteins are the anti-restriction proteins and the anti-CRISPRs. The anti-restriction proteins function via directly interacting with the restriction enzymes or masking the modification sites to cope with the RM system[10,11]. The mechanisms utilized by anti-CRISPRs to overcome host defense systems are diverse, encompassing the inhibition of nuclease activity, induction of non-specific DNA binding, prevention of target DNA binding, and various other strategies[12]. A recent study reported four distinct families of anti-defense proteins, including the anti-Gabija system protein Gad1 and Gad2, anti-Thoeris protein Tad2 and anti-Hachiman defense protein Had1[7]. The additional phage anti-defense mechanisms need more effort to investigate.

DSR2, a sirtuin (Sir2) domain-containing protein, demonstrates strong anti-phage defense capabilities through the abortive infection pathway[13]. Sir2 domain proteins are highly conserved from bacteria to

[1]MOE Key Laboratory of Rare Pediatric Diseases, Center for Medical Genetics, School of Life Sciences, Central South University, Changsha, Hunan, China. [2]Zhejiang Key Laboratory of Structural Biology, School of Life Sciences, Westlake University, Hangzhou, Zhejiang, China. [3]Westlake Laboratory of Life Sciences and Biomedicine, Hangzhou, Zhejiang, China. [4]State Key Laboratory of Developmental Biology of Freshwater Fish, Engineering Research Center of Polyploid Fish Reproduction and Breeding of the State Education Ministry, College of Life Sciences, Hunan Normal University, Changsha, Hunan, China. [5]These authors contributed equally: Ruiwen Wang, Qi Xu. ✉e-mail: rongyang@hunnu.edu.cn; shizhubing@westlake.edu.cn; chinalfx@163.com

human, which are shown to regulate various cellular processes in human, including chromatin remodeling, gene transcription, DNA repair and cell cycle[14]. In eukaryotes, the Sir2 domain proteins typically function as a deacetylase using NAD[+] as a cofactor[15]. While in bacteria, Sir2 domain proteins recently identified to act as NADases to hydrolyze the bacterial cellular NAD[+] to protect against phage infection. This phenomenon has been observed in multiple anti-phage systems, including Thoeris, SIR2-HerA, SIR2-APAZ/Ago and DSR systems[13,16–19]. Previous studies have revealed that DSR2 was activated by the newly translated TTP of SPR phage in the infected cells[13]. This activation triggers the degradation of NAD[+] in bacterial cells, thereby leading to bacterial cell death. Interestingly, the SPbeta phage has been found to evade DSR2-mediated anti-phage defense by expressing DSAD1, an inhibitor of DSR2. It was suggested that DSAD1 competes with TTP for DSR binding, effectively inhibiting the enzymatic activity of DSR2[13] However, the structure of DSR2 and the regulatory mechanism of DSR2 by TTP and DSAD1 necessitate further dedicated efforts for investigation.

In this study, we first biochemically characterized that DSR2 in solution assembled into an oligomer, and confirmed that the NAD[+] hydrolase activity could be triggered by TTP and shut down by the inhibitor protein DSAD1. Next, we solved the cryo-EM structures of apo DSR2, DSR2-TTP-NAD[+] and DSR2-DSAD1 complexes. These structures not only reveal the underlying mechanism of DSR2 oligomerization, activation and substrate NAD[+] coordination, but also illustrate the mechanism of DSR2 inhibition by DSAD1. In addition, we biochemically validated the molecular interfaces by mutagenesis assay. In summary, our findings provide a valuable template for comprehending the intricate mechanisms underlying bacterial anti-phage defense systems and phage immune evasion which holds great therapeutic potential for the development of phage-based therapies.

## Results

### Cryo-EM structure of DSR2 tetramer
DSR2, a Sir2-domain-containing NADase, can be activated by the phage TTP and suppressed by the SPbeta phage-encoded DSAD1 (Fig. 1a). To characterize the molecular mechanism of DSR2-mediated NAD[+] depletion, we first expressed and purified the *Bacillus subtilis* DSR2 protein, phage TTP and DSAD1. The in vitro NAD[+] hydrolase assay confirmed that the DSR2 can only be activated to degrade NAD[+] in the presence of TTP (Fig. 1b). DSR2 alone or in the presence of DSAD1 had no detectable NAD[+] hydrolase activity. The gel filtration profile shows that the DSR2 tends to form large oligomers (Supplementary Fig. 1a, b). To further investigate the mechanism of DSR2 oligomerization and NAD[+] binding, we collected the cryo-EM dataset of DSR2 H171A catalytically inactive mutant, and determined the cryo-EM structure of DSR2 (H171A) to overall 3.18 Å resolution (Supplementary Fig. 2 and Table 1).

The resolution of central part of DSR2 structure is better than 3 Å, and that of the very C-terminal region is relatively poor due to the dynamic property which was indicated by the local resolution maps and 3D variability analysis (Supplementary Fig. 2 and 3). The overall structure, especially the two ends, undergoes continuous motion (Supplementary Movie 1). Local refinement focused on the C-terminus of DSR2 improved the resolution, which allowed us to construct the atomic model and precisely assign the sidechain orientations, assisted by AlphaFold2 predicted model. The electron density map indicated that DSR2 is a tetramer (Fig. 1c, d). The protomer structure of DSR2 revealed an architecture that consists of an N-terminal Sir2 domain (residues 1–298) and a long C-terminal helical region (residues 299–1005). The Sir2 domains of two side-by-side DSR2 dimers contact each other to assemble into a head-to-head rod-shaped homo-tetramer of DSR2. The DSR2 Sir2 domain folds into a typical sirtuin-like domain, composed of 14 α-helices and 7 β-strands (Fig. 1e, f). The structure search and comparison revealed that the Sir2 domain of

DSR2 aligned remarkably well with the ADPR-bound Sir2 domain of ThsA (N112A) and the NAD[+]-bound Sir2 protein in the Sir2-HerA defense system, exhibiting the RMSDs of 3.2 Å and 10.2 Å, respectively[19,20] (Fig. 1g, h). The key residues constituting the catalytic pockets (N133, Y134, D135, and H171) in the Sir2 domain of DSR2 are remarkable conserved in the Sir2 protein of Sir2-HerA system and Sir2 domain of ThsA (Fig. 1i). The subsequent NADase assay unveiled that mutation of any of the key residues (Y134A, D135A or H171A) within the catalytic pockets obviously decreased the NAD[+] degradation activity in the presence of TTP (Fig. 1j). This structural feature suggests that DSR2, ThsA and Sir2 adopt the similar mode for NAD[+] binding and hydrolysis. The C-terminal helical region of DSR2 is composed of a helix-turn-helix (HTH) domain (residues 301–400), a helix bundle domain (residues 410–547) and a long HEAT repeat domain (residues 548–858) followed by a C-terminal domain (residues 859–1005) (Fig. 1k). The HTH domain folds into two helix-turn-helix (HTH) motifs with a β hairpin located between them. The HEAT repeat domain has six HEAT repeats (R1–R6). The HEAT domains of two DSR2 molecules in each end of DSR2 tetramer form two cavities with their C-terminal domains, which potentially interacts with its regulators (Fig. 1k). Interestingly, the two side-by-side DSR2 molecules form an asymmetric dimer, with the one cavity at DSR2 C-terminus closed while the other partially opened. Comparing these two cavities, we observed the structures of the C-terminal domains of two DSR2 protomers are distinct (Fig. 1l). The C-terminal domain with opened cavity rotates upward and thus moves away from the cavity center compared to the closed one.

### The molecular mechanism of DSR2 oligomerization
The overall structure of DSR2 tetramer indicates that the oligomerization of DSR2 is mainly mediated by four interfaces. The interfaces 1 to 3 are responsible for the DSR2 dimerization, while the interface 4 is critical for tetramerization (Fig. 2a).

The interface 1 consists of the contacts between the last two α-helices from the C-terminal domains of two adjacent DSR2 molecules (Fig. 2b). This interface is primarily maintained through hydrophobic interactions, wherein the hydrophobic side chains of I981, V988, L997, L1000, M1001, and L1005 from two DSR2 protomers form extensive hydrophobic interactions with one another (Fig. 2b). Interface 2 is formed within the HEAT repeat domains of the DSR2 helical region, where the interaction is mediated through a combination of hydrophobic and hydrophilic contacts. Specifically, the hydrophilic side chains of N548, Q549, and Q610 from one DSR2 protomer establish three hydrogen bonds with Q549', D553', and N563' residues from the neighboring protomer. Furthermore, the hydrophobic side chains of L558 and F559 from both adjacent protomers closely interact with each other, forming additional hydrophobic interactions that further stabilize the complex (Fig. 2c). Additionally, a long helix in R1 of the HEAT repeat domain of one DSR2 protomer with a partially opened cavity directly contacts the C-terminal domain from the other protomer, which facilitates the closure of its C-terminal cavity. Interface 3 is formed between the two adjacent Sir2 domains in the DSR2 dimer. Within this interface, hydrophilic interactions are observed between T206 and N202 residues from one DSR2 protomer with N202' and T206' residues from the adjacent protomer, respectively. Additionally, hydrophobic interactions are established between the hydrophobic side chains of W231 and L235 with P198' and L199' residues from the other molecule, respectively (Fig. 2d). The in vitro NADase assay revealed that mutations in key residues (L1000A/M1001A, N202A) on the dimerization interfaces of DSR2 significantly suppressed the activity of DSR2 triggered by TTP. These findings clearly indicate that dimerization of DSR2 is indispensable for its NAD[+] depleting activity (Fig. 2f).

Interface 4 plays a critical role in the tetramerization of DSR2 and is formed by the interaction between the Sir2 domains of two side-by-

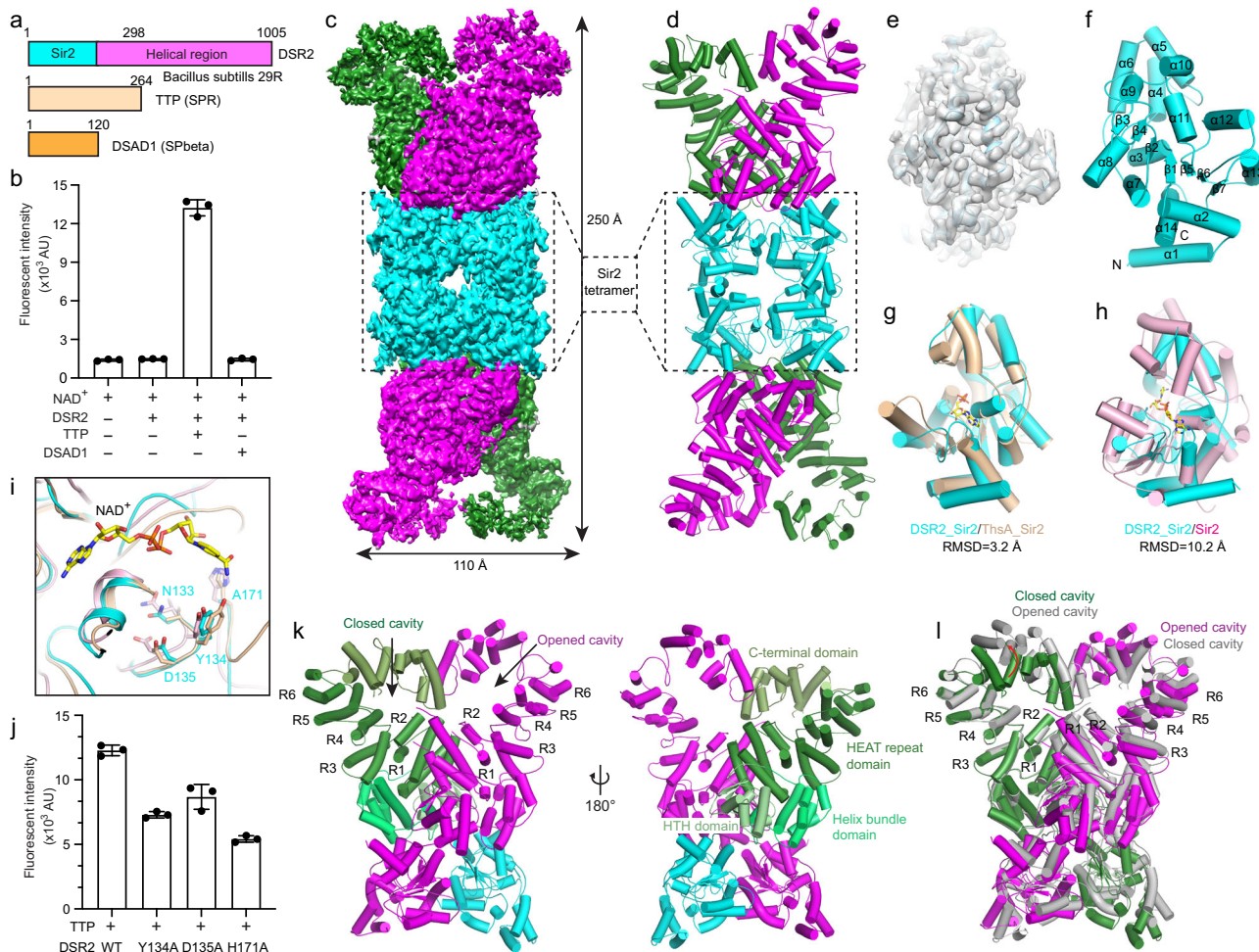

**Fig. 1 | DSR2 assembles into a rod-like head-to-head tetramer. a** Domain organization of DSR2 from *Bacillus subtilis* 29 R, activator TTP from SPR phage and inhibitor protein DSAD1 from SPbeta phage. **b** NAD⁺ hydrolase activities of DSR2 alone or in the presence of TTP and DSAD1. All experiments were replicated at least three times (mean ± SD, *n* = 3 independent replicates). **c** Cryo-EM density map of DSR2 tetramer in side view. The Sir2 domain tetramer is colored in cyan, while the helical repeats domain in the DSR2 dimer was colored in forest green and magenta. **d** Ribbon diagram of the DSR2 tetramer, and the domain was colored as in (**a**) and (**c**). **e** Electron density of the Sir2 domain of DSR2 docked with the Sir2 model. **f** Ribbon diagram of the Sir2 domain of DSR2. **g, h** Overlay of structures of the Sir2 domains of DSR2 and ADPR-bound ThsA N112A (PDB ID: 8BTP) or NAD⁺-bound Sir2-HerA system (PDB ID: 8UAF). DSR2, ThsA and Sir2 are colored in cyan, wheat and

light pink, respectively. **i** Close-up view of the NAD⁺-binding pockets of the Sir2 domains from DSR2, ThsA and Sir2-HerA system. The key residues for NAD⁺ coordination are shown in sticks. **j** NAD⁺ hydrolase activities of WT DSR2 or its mutants in the presence of TTP. All experiments were replicated at least three times (mean ± SD, *n* = 3 independent replicates). **k** Ribbon diagram of the DSR2 dimer in one side of the tetramer. The DSR2 helical region is divided into a helix-turn-helix (HTH) domain, a helix bundle domain, a long HEAT repeat domain and a C-terminal domain. Two partner-binding cavities (one opened cavity and one closed cavity) are constituted by the HEAT repeats and C-terminal domains of DSR2 dimer. **l** The opened cavity protomer of DSR2 aligned to the closed cavity promoter. Compared with the closed cavity, the helix of C-terminal domain with the opened cavity exhibits an outward twist.

side DSR2 dimers. Within this interface, the aromatic side chain of Y260 from one DSR2 protomer engages in hydrophobic interactions with I90' from the neighboring molecule, while the hydroxyl group on the Y260 side chain forms a hydrogen bond with Q89'. Moreover, two pairs of hydrogen bonds are established between Y261 and N226 residues with R86' from the other protomer (Fig. 2e). These interactions are also observed between H260', Y261', and N226' with I90, Q89, and R86, respectively, in the corresponding positions of the neighboring DSR2 protomers. Four such sets of interactions are formed in the Sir2 tetramer, facilitating the tetramerization of DSR2. To validate the role of this interface in maintaining DSR2 tetramerization, we generated DSR2 R86E and Y260E mutants and analyzed their oligomerization state using gel filtration and analytical ultracentrifugation (AUC). The gel filtration profiles of the DSR2 R86E and Y260E exhibited slower migration compared to the wild-type protein, indicating disruption of DSR2 tetramerization (Supplementary Fig. 4a). The molecular weights of wild-type (WT) DSR2 and its Y260E proteins, as

determined by AUC, were found to be 425 kDa and 215 kDa, respectively, confirming the disruption of DSR2 tetramerization caused by the Y260E mutation (Supplementary Fig. 4b). However, the AUC profile of DSR2 R86E exhibited a broad peak, and the calculated molecular weight was only 130 kDa, close to that of a monomer (Supplementary Fig. 4b). The enzymatic activity assay revealed that the R86E mutation dramatically reduced the activity of DSR2 (Supplementary Fig. 4c). These findings indicate that the R86E mutation likely impacts the folding of DSR2. Furthermore, the NADase activity of the DSR2 Y260E mutant exhibited no significant difference compared to the wild-type protein, indicating that tetramerization of DSR2 is dispensable for its activation (Fig. 2f).

**The cryo-EM structure of DSR2-TTP-NAD⁺ complex**
To investigate the molecular mechanism of DSR2 activated by TTP, we purified the DSR2 (H171A)-TTP complex protein and made the cryo-EM grids in the presence of NAD⁺ substrate, and its structure to an overall

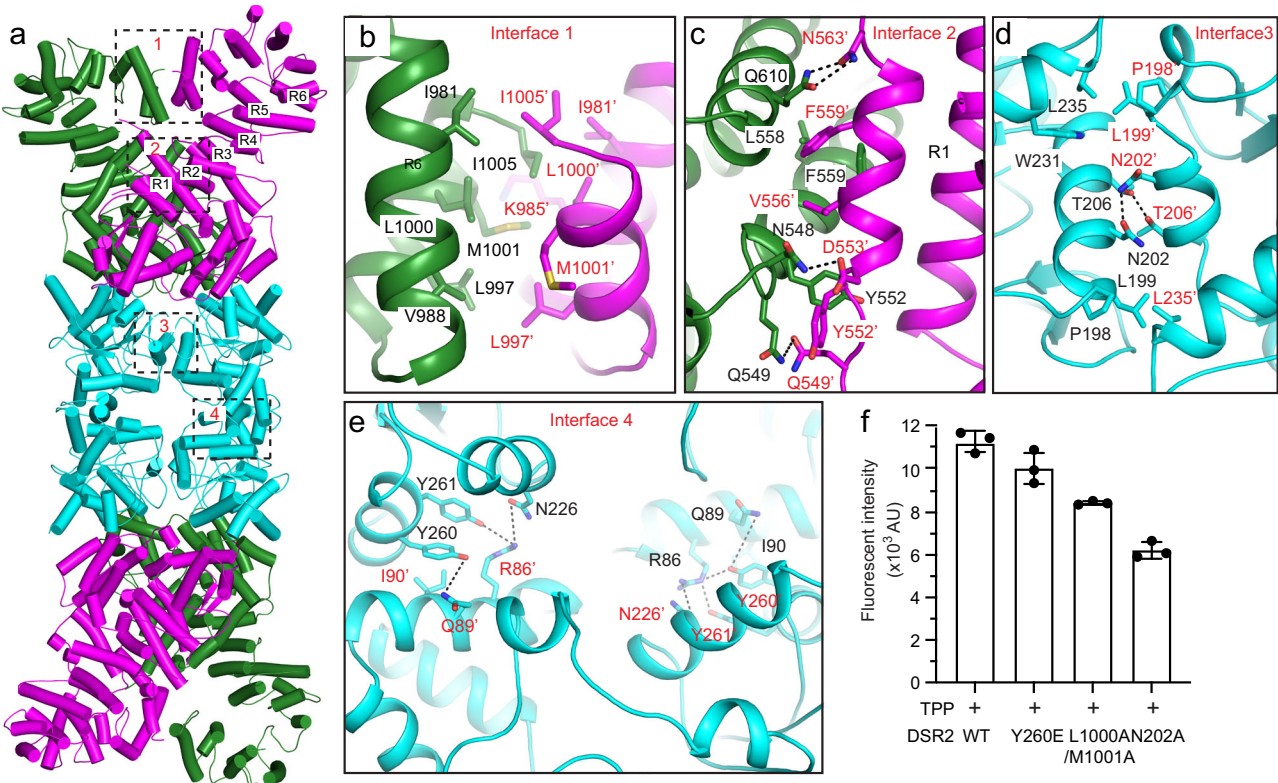

**Fig. 2 | The molecular interface of the DSR2 tetramer. a** Ribbon diagram of the DSR2 tetramer. The four main interfaces for DSR2 oligomerization are boxed by black dash lines. **b–d** Close-up views of the DSR2 dimer interfaces 1 (**b**), 2 (**c**) and 3 (**d**), with interacting residues are highlighted as sticks. The hydrogen bonds are indicated by the black dash line. **e** Close-up view of the DSR2 tetramer interface 4 consisted of Sir2 domain, with the key residues for interaction shown as sticks. **f** NAD$^+$ hydrolase activities of WT DSR2 or its mutants that disrupt the dimer or tetramer interfaces in the presence of TTP. In this assay, 1 μM of DSR2 WT or mutant proteins was preincubated with 8 μM TPP for 30 min. After the addition of 50 μM ε-NAD$^+$, the reaction was conducted at 37 °C for 15 min. The fluorescence intensity was measured using the Bio Tek Synergy H1 Plate Reader. All assays were performed at least in triplicate (mean ± SD, $n = 3$ independent replicates), and the standard deviations were calculated using GraphPad Prism.

resolution of 3.14 Å (Supplementary Fig. 5 and Table 1). Similar to the apo DSR2 structure, the resolution of C-terminal cavities region in the DSR2-TTP complex are still low due to the structural variability (Supplementary Movie 2). The local refinement focused on the TTP-binding region was employed to improve the resolution to 3.17 Å which made us to build the atomic model of TTP (Supplementary Fig. 6).

In the DSR2-TTP complex structure, DSR2 also folds into a head-to-head homo-tetramer. TTP asymmetrically binds to one of two adjacent DSR2 C-terminal cavities within the DSR2 dimer, forming a 2:4 hetero-hexamer (Fig. 3a, b). The electron density of TTP is missing from residues 80–167, while the remaining portion exhibits a β-sandwich structure consisting of one α-helix and ten β-strands (Fig. 3c). Compared to the apo DSR2 structure, four electron densities corresponding to NAD$^+$ were observed within the substrate-binding pocket of the Sir2 domain within the DSR2 tetramer (Fig. 3b, e). The NAD$^+$ molecules exhibit coordination through extensive hydrophobic and hydrophilic interactions. Specifically, the adenine and the nicotinamide groups of NAD$^+$ stack with Y282 and W60 of DSR2 to constitute two pairs of π-π interaction. In addition, the hydrophilic side chains of D249 and Q58 from DSR2 form two hydrogen bonds with the groups of adenine-ribose and nicotinamide-ribose, respectively. Furthermore, the hydroxyl group of DSR2 Y84 forms close contacts with the phosphate and nicotinamide groups, resulting in the formation of two pairs of hydrogen bonds (Fig. 3d). The superimposed structures of the Sir2 domains of DSR2 (NAD$^+$-bound) and ThsA (ADPR-bound) revealed that the adenine group occupies the same pockets in both Sir2 domains, while the nicotinamide groups exhibit differences (Fig. 3f). In the ThsA structure, the phosphate group of ADPR points into the catalytic pocket formed by the residues

N112, Y113, D114, and H152, which correspond to N133, Y134, D135, and H171 in DSR2. These residues, N112 and H152 in ThsA, constitute the catalytic dyad. However, in the DSR2 Sir2 structure, the nicotinamide group deviates from this catalytic pocket (Fig. 3f). Considering that the residues Y134, D135, and H171 are essential for the hydrolase activity of DSR2, the DSR2-TTP-NAD$^+$ complex structure may represent an intermediate state during NAD$^+$ degradation.

One TTP molecule simultaneously bind two adjacent DSR2 protomers. Their interaction is predominantly facilitated by three pairs of intermolecular anti-parallel β-sheets (Fig. 3g). Specifically, the β1 (residues K2 to Q6) and β4 (residues S25 to Q28) strands of TTP form two anti-parallel β-sheets with two β strands of the neighboring DSR2 protomer, one of which is between the HTH and helix bundle domains (residues L403′ to I407′) and the other is in the HEAT repeat domain (residues S573′ to F576′) (Fig. 3g). Furthermore, the β10 strand of TTP, encompassing residues A231 to I234, constitutes a third β-sheet with a β strand between residues I904 and F907 in the C-terminal domain of the DSR2 protomer, where TTP is predominantly situated (Fig. 3g). This β strand of DSR2 C-terminal domain is disordered in DSR2 in apo and DSAD1-bound structures (see below). The existing interfaces between DSR2 and TTP indicate that dimerization of DSR2 is required for binding to TTP. To clarify the molecular mechanism of the asymmetric binding of TTP to DSR2, we performed a structural superimposition of the C-terminal cavities of the DSR2 dimer in both TTP-bound and unbound forms. The superimposed structures reveal that the interaction between TTP and DSR2 induces a conformational rearrangement of the C-terminal region, leading to a more compact and closed conformation (Fig. 3h).

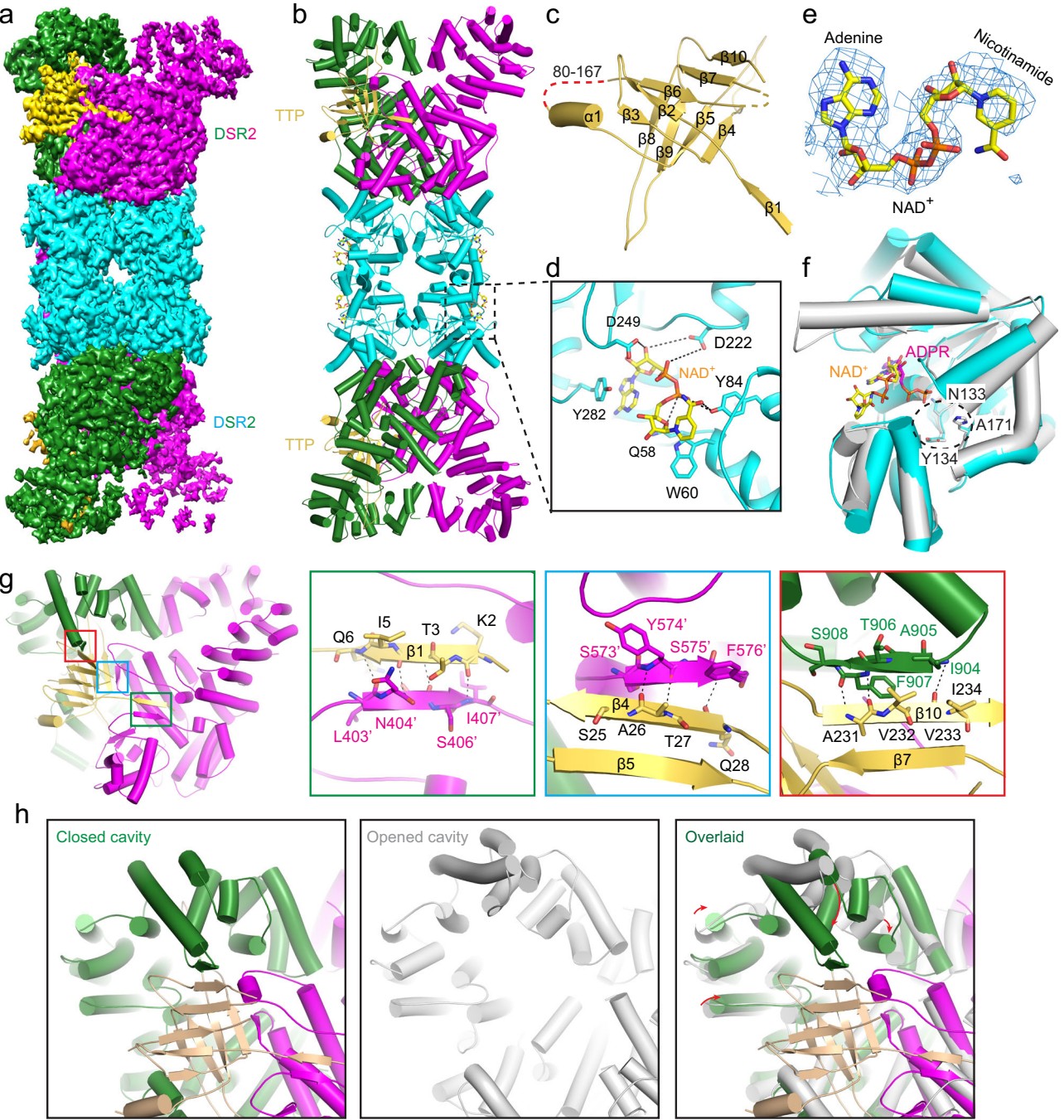

**Fig. 3 | The cryo-EM structure of DSR2-TTP-NAD+ complex. a** Cryo-EM density maps of DSR2-TTP-NAD+ complex. In this drawing, the Sir2 domain tetramer is colored in cyan, while the helical repeats domain in the DSR2 dimer was colored in forest green and magenta and TTP colored in wheat. **b** Ribbon diagram of the DSR2-TTP-NAD+ complex structure, and the domains were colored as in (**a**). **c** Ribbon diagram of TTP, the secondary structures were labeled and the missing residues in the cryo-EM structure were indicated by the red dash line. **d** Close-up view of the NAD+ binding pocket in the DSR2 Sir2 domain, with the key residues for the NAD+ coordination shown as sticks. **e** The electron density map of NAD+ observed in the cryo-EM structure. The adenine and nicotinamide groups of NAD+ was labeled. **f** Superimposed structures of the Sir2 domains of DSR2 (NAD+-bound) and ThsA (ADPR-bound). The catalytic pocket composed by the H171, Y134 and N133 is indicated by the black dashed cycle. **g** Close-up view of the interaction surface between DSR2 and TTP. Three pairs of intermolecular anti-parallel β-sheets are formed between DSR2 and TTP. **h** Closed-up views of DSR2 closed cavity bound with TTP, opened cavity and superimposed structures of opened cavity to TTP bound cavity.

## The overall structure of DSR2-DSAD1 complex

The bacteria phage SPbeta expressed DSAD1 protein to inhibit the NADase activity of DSR2 for immune evasion[13]. To elucidate the molecular mechanism underlying the specific interaction between DSAD1 and DSR2, we initially co-expressed DSAD1 and DSR2 in *E. coli* and subsequently co-purified the complex proteins (Supplementary Fig. 1c, d). Then, we collected the cryo-EM datasets of DSR2-DSAD1 complex for structural analysis. The resulting 3D reconstruction achieved an overall resolution of 2.9 Å (Supplementary Fig. 7 and Table 2). Similar to the apo DSR2 structure, the electron density within the central region of the DSR2-DSAD1 complex exhibited high quality, with a resolution exceeding 3 Å. In comparison with that of DSR2 alone, the map of the DSR2-DSAD1 complex displays an extra density on the C-terminal cavity of DSR2,

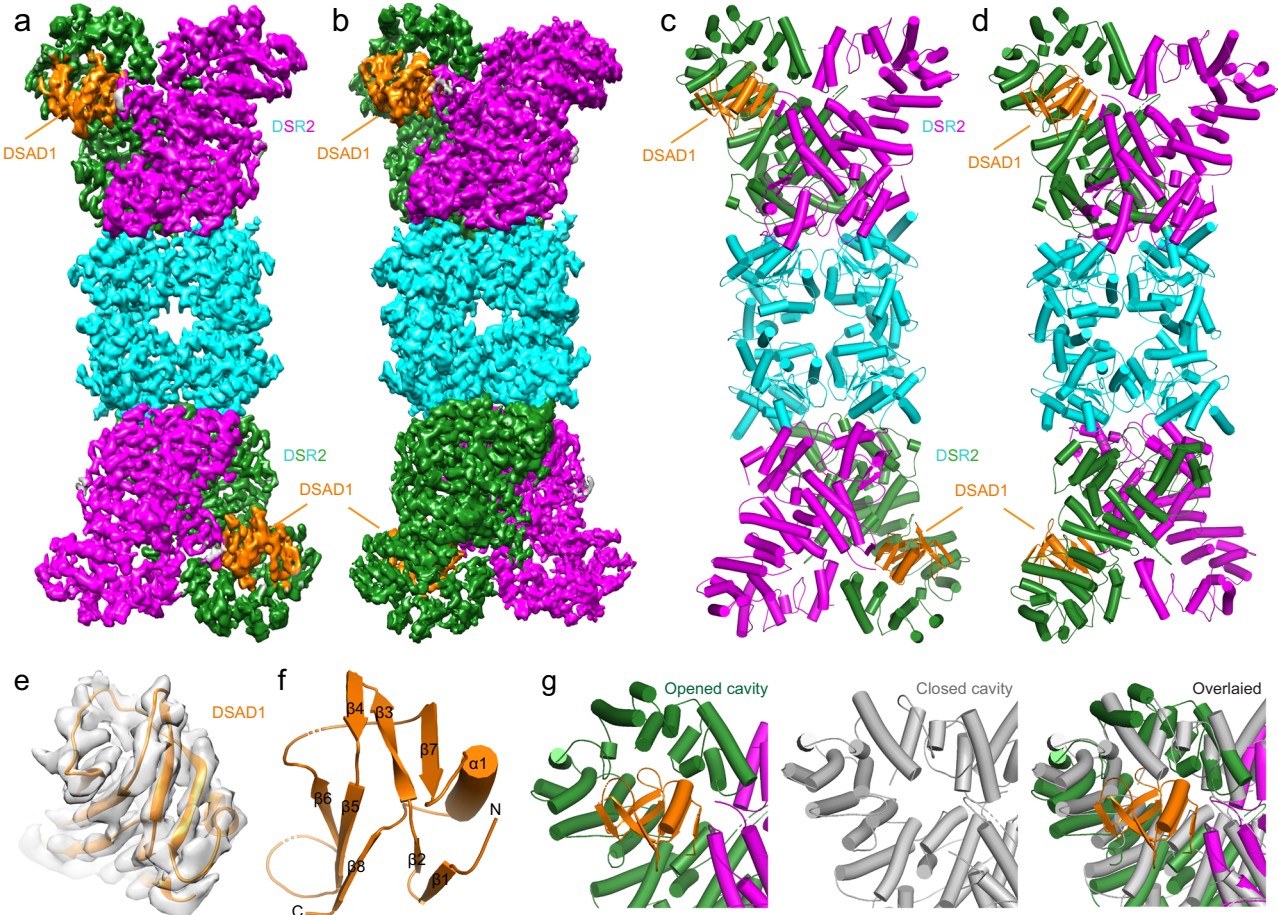

**Fig. 4 | The cryo-EM structure of DSR2-DSAD1 complex. a, b** Cryo-EM density maps of DSR2-DSAD1 complex with DSAD1 on the same side (**a**) or opposite sides (**b**). The Sir2 tetramer is colored in cyan, while the helical region in the DSR2 dimer is colored in forest green and magenta, and the DSAD1 is colored in orange. **c, d** Ribbon diagram of the structures of DSR2 bound with two DSAD1 molecules on the same side (**c**) or opposite sides (**d**). The domains are colored as above. **e** Electron density of the DSAD1 docked with the structural model. **f** Ribbon diagram of the DSAD1. **g** Closed-up views of DSR2 opened cavity bound with DSAD1, closed cavity and superimposed structures of closed cavity to the DSAD1 bound cavity.

which corresponds to DSAD1. However, the resolution of DSAD1 and its binding region in DSR2 is relatively low, posing challenges for model building (Supplementary Fig. 8 and Supplementary Movie 3). By employing local refinement focused solely on the DSR2-DSAD1 interaction region, the resolution for this specific area was improved to 3.2 Å. The sidechains of the majority of DSAD1 residues were distinctly visible, enabling successful de novo construction of the atomic model for DSAD1 and the entire DSR2-DSAD1 complex (Fig. 4e and Supplementary Fig. 8 and Table 2). The DSAD1 structure comprises one α-helix and eight β-strands, forming three intertwined β sheets, which deviates significantly from the initially predicted AlphaFold2 model (Fig. 4f).

The cryo-EM structure of DSR2-DSAD1 complex unveiled that DSR2 in the complex also forms a head-to-head tetramer. Interestingly, two DSAD1 molecules were observed to bind to the C-terminal regions of DSR2 on the same side or opposite sides of the tetramer, forming a complex with 2:4 molar ratio (Fig. 4a–d). That is to say, the two adjacent DSR2 C-terminal cavities within each DSR2 dimer exhibit the ability to bind only one DSAD1 molecule (Fig. 4a, b). This is caused by the asymmetric property of DSR2 dimer at each end as mentioned above (Fig. 1i). DSAD1 only binds to the partially opened cavity but not the closed one. Comparing the two cavities of DSR2 dimer, we found the closed one is too small to accommodate the DSAD1, and the conformational discrepancy of C-terminal domain mentioned above also prevents from DSAD1 binding (Fig. 4g).

## The structural analysis and validation of the interaction between DSR2 and DSAD1

The surface view of the DSR2-DSAD1 complex structure revealed that DSAD1 engages in simultaneous interactions with two adjacent DSR2 protomers, and the primary interaction region can be divided into three distinct interfaces. The interfaces 1 and 2 are constituted by the DSAD1 with the HEAT repeats and C-terminal domain of DSR2 that form the partner-binding cavity, while the third interface is established with the HEAT repeats from the neighboring DSR2 protomer (Fig. 5a).

In the interface 1, a long flexible loop between β5 and β6 of DSAD1 is embedded into the cavity of DSR2, while in the interface 2, the body of DSAD1 locates on the top of the cavity. The interaction in the interface 1 is mainly mediated by the hydrophobic contacts. The aromatic sidechain of F59 from DSAD1 inserts into a hydrophobic pocket constituted by the sidechains of I918, W919, L922 and L966 from DSR2 (Fig. 5b). Moreover, three pairs of hydrogen bonds are observed in the interface 2 to further stabilize the complex. The side chain of N21 from the DSAD1 establishes two hydrogen bonds with the nearby sidechains of S957 and K960 from DSR2. The formation of the third hydrogen bond involves the side chains of S18 from DSAD1 and of D993 from DSR2 (Fig. 5c). The contacts between DSAD1 and the neighboring DSR2 protomer contribute to the interface 3. The first helix of the HEAT repeat domain and the following repeat R1 directly contact DSAD1. This region contributes a β strand that is disordered in apo structure to

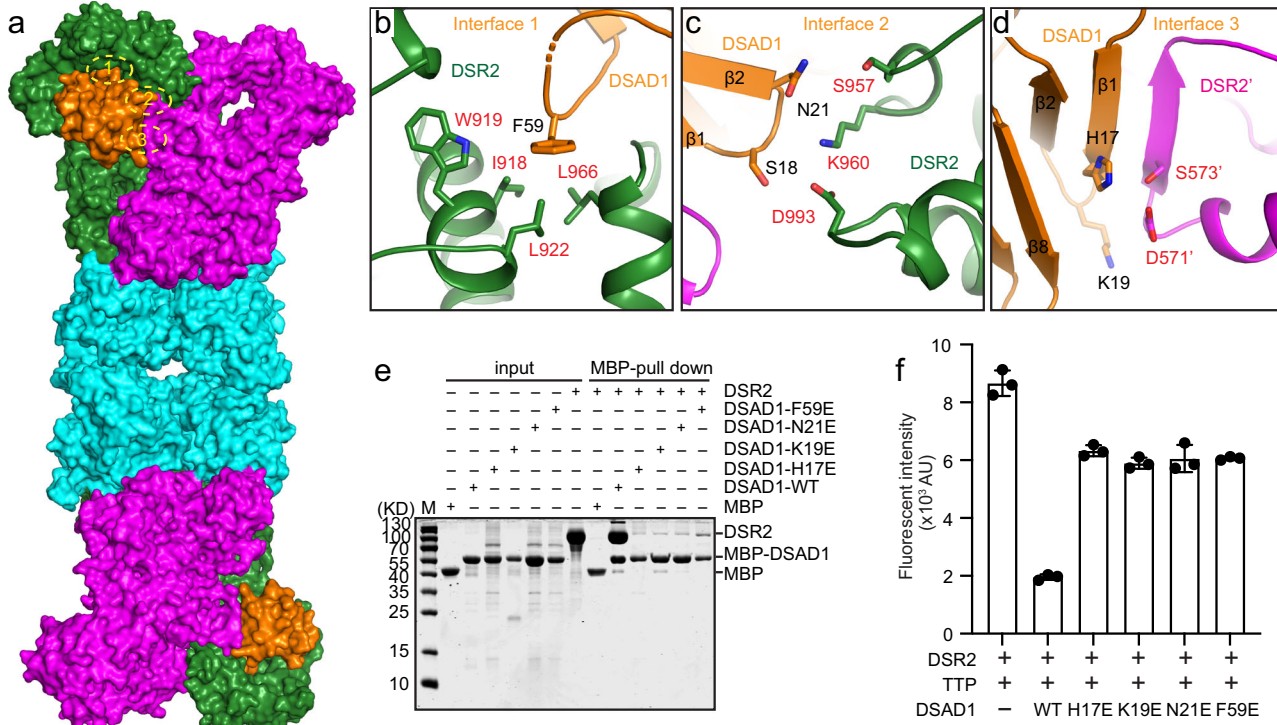

**Fig. 5 | The interface analysis and validation of the DSR2-DSAD1 complex.**
**a** Surface view of DSR2-DSAD1 complex with DSAD1 on the same side. **b**–**d** Close-up views of the interfaces between DSAD1 and DSR2. The interface 1 (**b**) and 2 (**c**) are established by DSAD1 and the DSR2 protomer that constitute the partner-binding cavity, while the interface 3 (**d**) is formed by DSAD1 with the neighboring protomer of the DSR2 dimer. An inter-molecular anti-parallel beta sheet is established between DSAD1 and DSR2 in the interface 3. **e** MBP pull-down assays to analyze the binding of the wide-type DSAD1 and its mutant proteins to DSR2. The experiment was independently replicated three times, yielding similar results. **f** In vitro NAD⁺ hydrolase assay was employed to analysis the activity of DSR2-TTP complex inhibited by WT DSAD1 or its mutants. In this assay, 1 μM DSR2 proteins was pre-mixed with 4 μM DSAD1 proteins, and then, 8 μM TTP was added into the mixtures after 1 h incubation. Finally, the NAD⁺ hydrolase activity of the protein complex was measured according to the standard process. All experiments were replicated at least three times (mean ± SD, $n = 3$ independent replicates).

form a three stranded β sheet with β1 and β2 of DSAD1, rendering the partially opened cavity closed (Fig. 5d). The polar side chains of K19 and H17 in DSAD1 are positioned in close proximity to the hydrophilic side chains of D571' and S573' from DSR2. This proximity suggests the potential for the establishment of two hydrogen bonds, which could effectively facilitate the formation of the complex (Fig. 5d). To further validate the interfaces, we generated mutants on DSAD1 based on the three identified interfaces and then analyzed binding affinities of DSAD1 mutants to DSR2 by in vitro pulldown assay. The results of the pulldown assay demonstrated that any single mutation in DSAD1 within each interface, such as F59E in interface 1, N21E in interface 2, and H17E or K19E in interface 3, significantly reduced the binding affinity of DSAD1 to DSR2 when compared to the wild-type DSAD1 proteins (Fig. 5e). The NADase assay revealed that these DSAD1 mutants abolished the inhibition ability to the activity of DSR2 triggered by TTP (Fig. 5f).

**The competitive binding of DSAD1 and TTP to DSR2 effectively inhibits its NADase activity**
The structures of DSR2-DSAD1 and DSR2-TTP complexes revealed that both DSAD1 and TTP bind to C-terminal cavity of DSR2, which implied the direct competition between DSAD1 and TTP for binding to DSR2. To validate the competition mechanism, we first performed the in vitro NADase assay. TTP stimulated the activity of DSR2 in a dose-dependent manner. However, the purified DSR2-DSAD1 complex showed limited activation by TTP (Fig. 6a). Conversely, the NAD⁺ hydrolase activity of the DSR2-TTP complex was progressively suppressed with increased concentrations of DSAD1 (Fig. 6b). In addition, the Strep pulldown results indicated that the direct binding of DSR2 to TTP could be

gradually inhibited with increased amounts of DSAD1 (Fig. 6c). These findings provide compelling evidence that DSAD1 effectively inhibits the NADase activity of DSR2 by competing with TTP for binding to DSR2.

**TTP-triggered structural rearrangement activate NADase activity of DSR2**
To explore the molecular mechanism underlying the activation of NADase activity of DSR2 by TTP, we superimposed the structures of DSR2 in apo, TTP-bound and DSAD1-bound states and conducted an analysis of the resulting structural variations (Fig. 6). We did not observe dramatic conformational changes in the DSR2 Sir2 domain upon the binding of TTP and DSAD1. The key residues W60, Y84 and Y282 which are responsible for NAD⁺ coordination, as well as N133, Y134 and H171 located in the catalytic center, only undergo slightly displacement from their original positions within the Sir2 domain (Fig. 6d). However, the remarkable structural variation was observed on the C-terminal partner-binding cavities of DSR2 upon TTP binding. A small angle of inward twist was detected at its C-terminal region to assist TTP binding (Fig. 6d). Although no dramatic conformational changes occur in Sir2 domain upon TTP binding, we observed a large motion in the HTH domain of DSR2 in TTP-bound state compared to apo and DSAD1-bound states. This motion only occurs on the TTP-binding side in the DSR2-TTP structure. TTP, but not DSAD1, directly contacts DSR2 HTH domain (Fig. 6d). As the HTH domain of DSR2 is linked to and also contacts the Sir2 domain, the motion occurring in the HTH domain would transmit to the Sir2 domain and influence its conformation, which could activate DSR2. The importance of the interface between the HTH domain and the Sir2 domain in the

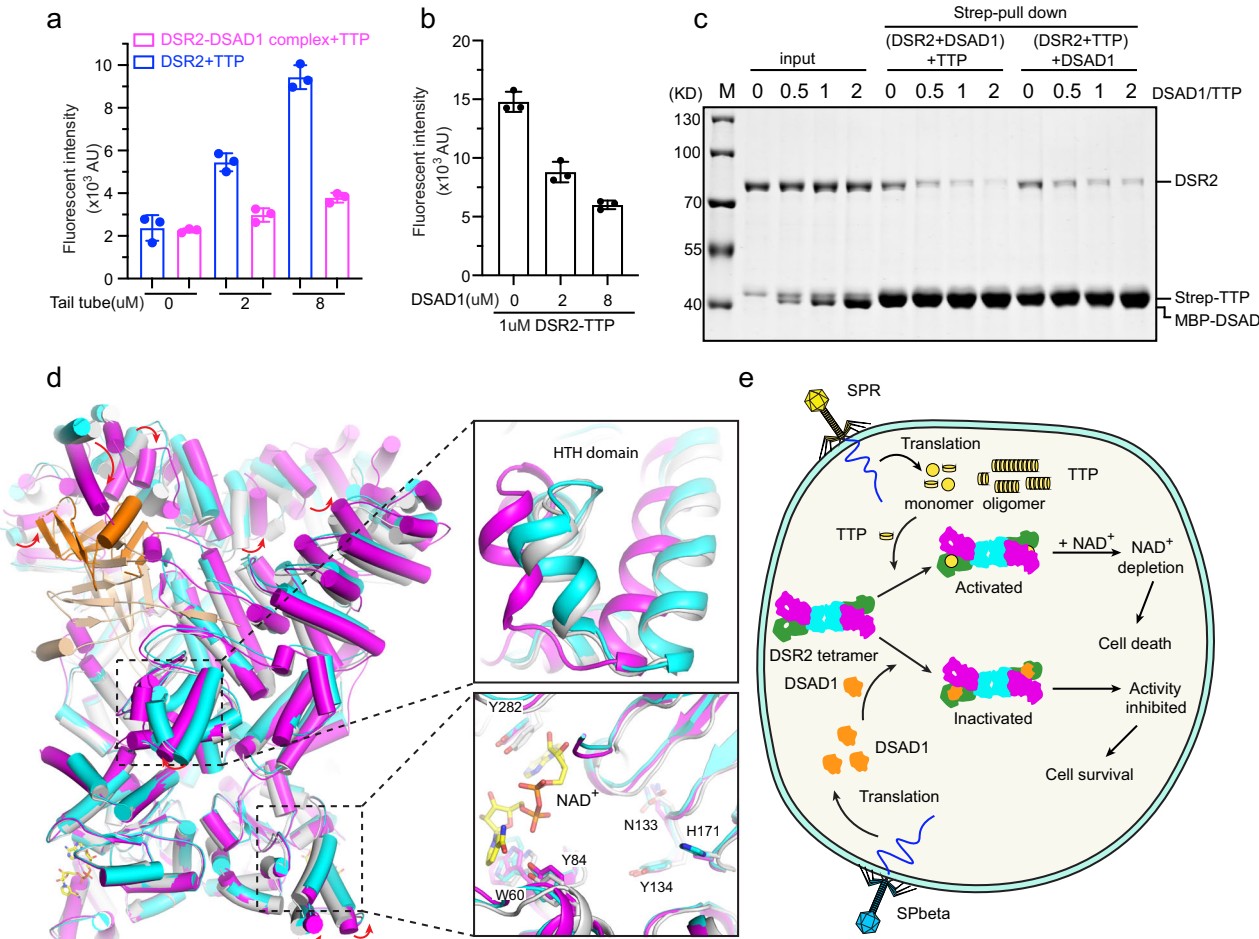

**Fig. 6 | The competitive binding of DSAD1 and TTP to DSR2 effectively inhibits its NADase activity. a** NAD+ hydrolase activities of DSR2 or DSR2-DSAD1 complex in the presence of different amount of TTP. All assays were performed at least in triplicate (mean ± SD, $n = 3$ independent replicates), and the standard deviations were calculated using GraphPad Prism. **b** NAD+ hydrolase activities of DSR2-TTP complex in the presence of different amount of DSAD1 protein. All experiments were replicated at least three times (mean ± SD, $n = 3$ independent replicates). **c** Strep pull down assay was performed to analyze the direct competition between DSAD1 and TTP for binding to DSR2. In this assay, DSR2 was pre-incubated with different amount of DSAD1 or TTP for 1 h, and then, the third protein was added to the reaction mixtures for evaluate the competitive binding to DSR2. The experiment was replicated independently three times, producing similar results. **d** Superimposed ribbon diagrams of the DAR2 apo, DSR2-TTP and DSR2-DSAD1 complex. DSR2 dimers in the DSR2-DSAD1 complex, DSR2-TTP complex and apo DSR2 are colored in cyan, magenta and gray, respectively. TTP and DSAD1 are colored in wheat and orange, respectively. The key residues required for NAD+ coordination but shifted relative to the other protomer are shown as sticks. **e** A proposed model to illustrate the mechanisms of DSR2 activation upon phages infection and phages immune evasion against the DSR2-dependent defense system.

activation of DSR2 is further validated by the three recently published articles[21–23].

It is unexpected that no obvious structural rearrangement in DSR2 Sir2 domain upon TTP binding given that TTP is an activator of DSR2. The possible reason is that the structure that we captured is still not in fully active state as we used the catalytically inactive mutant, H171A. This is consistent with the observation that the substrate NAD+ does not position at the catalytic site. It is possible that the conformation would change upon NAD+ contacting the catalytic residues in DSR2 Sir2 domain including N133 and H171. Further studies on the structure of the DSR2-TTP-NAD+ complex in its fully active state will provide deeper insights into the activation mechanism of DSR2. Although the conformation of the C-terminal cavities of DSR2 in the DSR2-DSAD1 complex has some variations compared to the DSR2-apo structure, the HTH domain of DSR2 in the DSR2-DSAD1 complex and the DSR2-apo structure shows no obvious variation. This suggests that the inhibitor DSAD1 locks DSR2 in an inactive state, preventing its activation.

Based on our results, we proposed a model to illustrate the mechanisms of DSR2 activation upon phages infection and phages immune evasion against the DSR2-dependent defense system (Fig. 6e).

In this model, DSR2 is expressed and assembles into a homo-tetramer, remaining in an inactive state prior to phage infection. Upon infection, the phage DNA is inserted into the bacterial cell and phage proteins are translated. The newly translated monomer form of phage TTP binds to the C-terminal partner-binding cavities of DSR2 tetramer, activating its NADase activity. This activation triggers bacterial cell death, inhibiting phage propagation by depleting cellular NAD+. However, a family of SPbeta phages has evolved to escape the DSR2-dependent immune system by expressing the inhibitor protein DSAD1. DSAD1 competes with TTP for DSR2 binding, effectively shutting down the NADase activity of DSR2 and allowing the bacterial cells to survive. Consequently, the SPbeta phage can complete its reproduction through the bacterial translation system.

## Discussion

Numerous evidence proved that NAD+ is an essential central metabolite for survival which acts as coenzymes participating in diverse cellular physiological processes, including glycolysis, the Krebs cycle, and oxidative phosphorylation and some additional processes[24]. NAD+ also functions as cofactors for Sir2-domain containing deacetylase and

PARPs to catalyze the deacetylation and PARylation of the substrates[15,25]. In prokaryote, bacteria evolved a kind of anti-phage defense system via depleting the cellular NAD$^+$ to trigger the bacterial cell death, thereby inhibiting the phage propagation[3]. The TIR and Sir2 domain proteins shown to have NAD$^+$ hydrolysis activity holding the central role in multiple anti-phage systems, including CBASS, Thoeris, SIR2-HerA, Sir2-APAZ/Ago, TIR-APAZ/Ago, DSRs, and some other systems[16,18,19,26,27]. The key question of these NADase-based anti-phage system is how to accept the infection signals to precisely control the enzymatic activity. In the TIR-APAZ and Sir2-APAZ systems, the NADase activity of TIR and Sir2 are both activated by binding to the target DNA[17,28,29]. In the CBASS system, TIR-STING receives signals through STING domain binding to cyclic dinucleotides, leading to filament assembly and NAD$^+$ depletion[30]. However, the mechanisms by which bacteria distinguish between their own DNA and phage DNA, as well as the activation of bacteria cGAS to synthesize signaling molecules during phage infection, remains elusive.

The mechanism of DSR2-mediated anti-phage system sensing the infection signal has recently been demonstrated. DSR2 is activated by directly recognition of the newly translated TTP of SPR phage[13]. In this study, we initially confirmed the regulation of enzymatic activity of DSR2 by TTP and DSAD1 using in vitro NADase assay, then determined the cryo-EM structure of apo DSR2 and DSR2-TTP-NAD$^+$ complex. DSR2 in our structures arrange as a dimer of dimers mediated by its N-terminal Sir2 domains. The oligomeric state is commonly observed in the other NADase domains. For instance, the TIR domain in TIR-APAZ and TIR-STING systems forms a tetramer and an oligomeric filament, respectively[28,30]. In the Sir2-HerA system, the Sir2 domain exhibits a dodecameric architecture[18,19]. The Sir2-containing NADase ThsA in the Thoeris defense system is activated through binding to 1"-3' gcADPR, thereby triggering the assembly of helical filaments[20]. Oligomerization appears to be a common property among NADase domains, potentially contributing to their activation. Indeed, our NADase assays confirmed that the dimerization of DSR2 is critical for its enzymatic activity. Interestingly, even though the Y260E mutant disrupts the tetramerization, the NAD$^+$ hydrolase activity of DSR2 remained unaffected, suggesting the tetramerization is not essential for its activity (Fig. 2f). The Sir2 domain of DSR2 adopts a canonical sirtuin-like domain fold, exhibiting good alignment with the Sir2 domain of ThsA in the Thoeris system and the Sir2 protein in the Sir2-HerA system, implying a similar mechanism for NAD$^+$ coordination and hydrolysis. This observation suggests that the lack of the activator protein might have resulted in DSR2 being in an inhibited state.

To cope with the diverse anti-phage defense systems of bacteria, phages evolved to encode anti-defense proteins and non-coding RNAs for immune evasion[7,31,32]. DSAD1 protein encoded in SPbeta phage was discovered to be an effective inhibitor of DSR2, which can inactivate the NADase activity of DSR2 by competing with TTP for direct binding. We have solved the high-resolution structure of the DSAD1-DSR2 complex, which unveils an asymmetric binding of DSAD1 to DSR2. Both DSAD1 and TTP bind to C-terminal cavity of DSR2, which implied the direct competition between DSAD1 and TTP for binding to DSR2. Actually, our biochemical results indicate that DSAD1 can displace TTP from DSR2 with higher concentration, but not vice verse (Fig. 6c). Although we don't know the expression order of the activator and inhibitor, the NADase assays conducted by mixing DSR2 with TTP and DSAD1 in different orders clearly demonstrate that DSAD1 can effectively suppress the activity of DSR2 regardless of whether DSR2 encounters TTP or DSAD1 first (Fig. 6a, b). Furthermore, the superimposed structural analysis demonstrates that contact with DSAD1 induces a conformational change in the overall DSR2 structure, which may lock the DSR2 in the inactivated state.

Our structural studies have revealed the molecular mechanisms underlying the DSR2-mediated anti-phage system in bacteria and DSAD1-dependent phage immune evasion. These findings hold the

potential to guide artificial modifications on bacterial phages that can selectively target and eliminate pathogenic bacteria, thereby paving the way for the development of innovative therapies to combat multidrug-resistant bacterial infections.

## Methods

### Protein expression and purification

The cDNAs encoding the full-length DSR2 from *Bacillus subtilis* 29 R (WP_029317421), DSAD1 from SPbeta phage (WP_004399562) and TTP from SPR phage (WP_010328117) were synthesized from Tsingke company. The DNA fragments encoding the full-length DSR2, DSAD1 and TTP were amplified by PCR and cloned into the pRSF-32M vector (a modified version of the pRSF-Duet vector that introduces N-terminal thioredoxin and 6xHis tags before the first multiple cloning site) and the pET-32M.3 C vector (a modified version of the pET-32a vector that introduces N-terminal thioredoxin and 6xHis tags before the first multiple cloning site) for the subsequent recombinant protein expression. To prepare the MBP-tagged recombinant DSAD1 proteins for pull-down assay, the DNA fragment encoding the DSAD1 were cloned into the pET-MBP vector (a modified version of the pET-32a vector that introduces a N-terminal Maltose Binding Protein before the first multiple cloning site). The construction of plasmids for this study was performed using a Seamless Assembly Kit following the manufacturer's protocol (ABclonal, RK21020). All point mutations in DSAD1 and DSR2 utilized in this study were generated using the standard PCR-based mutagenesis method and were further validated by DNA sequencing. The Sequences of oligonucleotides (primers) are listed in the Supplementary Data.

The recombinant proteins of DSR2, MBP-DSAD1, TTP and DSR2-DSAD1 complex were expressed in BL21 (DE3) *Escherichia coli* cells. The bacterial were cultured in LB medium at 37 °C. Once the OD$_{600}$ of the culture reached 0.8, the temperature was reduced to 16 °C, and protein expression was induced by adding 200 μM IPTG. The cultures were then incubated overnight at 16 °C to allow for further protein expression.

To purify the DSR2 and TTP, bacterial cell pellets were collected by centrifugation at 3000 g for 10 min. The pellets were subsequently resuspended in five volumes of binding buffer (50 mM Tris-HCl, pH 7.9, 500 mM NaCl, 5 mM imidazole). The bacterial cells were then lysed using an ultrahigh-pressure homogenizer machine (ATS-1500, ATS Engineering Limited). The cell lysate was centrifuged at 35,000 g and 4 °C for 30 min to remove the cellular debris. The resulting supernatant was carefully transferred to a new 50-ml centrifuge tube and mixed with pre-equilibrated Ni$^{2+}$-NTA agarose resin (Qiagen, 30230) by the binding buffer. The mixture was incubated at 4 °C for 1 h with rotation. Following extensive washing with wash buffer (50 mM Tris-HCl, pH 7.9, 500 mM NaCl, 30 mM imidazole), the 6 x His-tagged proteins were eluted from the Ni$^{2+}$-NTA resin using elution buffer (50 mM Tris-HCl, pH 7.9, 500 mM NaCl, 400 mM imidazole). The target proteins were subsequently concentrated, filtered, and further purified using size exclusion chromatography with a UNIONDEX 200PG column (Union Biotech). The peak fraction proteins were analyzed by SDS-PAGE and stained by Coomassie Brilliant Blue. The fractions with pure proteins were combined for concentration, and then stored at -80 °C for future use.

For the preparation of the DSR2-TTP complex, the DSR2 (H171A) and TTP were separately expressed in BL21 codonlus (DE3) RIL *Escherichia coli* cells. After 16 h induction with IPTG at 16 °C, the cell pellets containing the target proteins were collected by centrifugation and then directly mixed together. The complex proteins were initially purified by Ni$^{2+}$-NTA affinity chromatography according to the standard process. The eluted proteins were further purified by the Resource Q column and gel filtration size extrusion. The peak fractions were collected for concentration, and then stored at -80 °C for future use.

For the purification of DSR2-DSAD1 complexes, the plasmids containing the DSR2 and DSAD1 genes were co-transformed into BL21 (DE3) *Escherichia coli* bacterial cells. The cell culture and protein expression were carried out using standard procedures. The protein purification process followed the same steps as the apo DSR2 purification.

To purify the MBP-tagged proteins, bacteria pellets were collected and resuspended in five volumes of PBS buffer. The cells were lysed by the ultrahigh-pressure homogenizer. The mixture was cleared by centrifugation at 35,000 g and 4 °C for 30 min. The supernatant was transferred into a new 50 ml centrifuge tube and mixed with appropriate amylose resin (NEB, E8021L) that had been pre-equilibrated by the PBS buffer. The mixture was rotated at 4 °C for 1 h for incubation. After extensive wash with PBS buffer, the MBP-tagged proteins were eluted from the resin by 15 mM D-Maltose dissolved in the buffer containing 50 mM Tris-HCl, pH 7.5, 200 mM NaCl, 1 mM DTT. The proteins were further purified by the UNIONDEX 200PG column (Union Biotech). The peak fraction proteins were pooled, concentrated, and stored at -80 °C.

## Analytical ultracentrifugation (AUC) assay

Sedimentation velocity experiments were carried out on an Optima XL-1 analytical ultracentrifuge (Beckman Coulter) equipped with a four-cell rotor at 12 °C. Samples were diluted to 0.8 mg/ml in the gel filtration buffer containing 20 mM Tris-HCl, pH 7.5, 100 mM NaCl, and 1 mM DTT. The sedimentation velocity data were calculated with the software SEDFIT[33].

## In vitro NAD$^+$ hydrolase assay

ε-NAD$^+$ assays were performed in a 150 µl reaction system containing 1 µM DSR2-TTP, 50 µM ε-NAD$^+$ (Sigma-Aldrich) and the indicated amounts of DSAD1 proteins. The WT or mutants DSR2 protein were pre-incubate with TTP in the reaction buffer (10 mM MES, pH 6.5, 150 mM NaCl and 5 mM MgCl$_2$) on ice for 30 min. For inhibition assay, the WT or mutants proteins of DSAD1 were added into the reaction mixtures and incubated on ice for an additional 30 min. After the addition of 50 µM of ε-NAD$^+$, the reaction was conducted at 37 °C for 15 min and the 96-well plate was transferred to pre-heated Bio Tek Synergy H1 Plate Reader. The fluorescence intensity was measured with the an excitation of 310 nm and an emission wavelength of 410 nm. All assays were performed in triplicate, and the standard deviations were calculated using GraphPad Prism.

## MBP pulldown assay

The interaction between DSR2 and MBP-tagged DSAD1 proteins were analyzed in the reaction buffer containing 25 mM Tris, pH 7.5, 100 mM NaCl, 1 mM DTT. 50 µg of MBP-tagged DSAD1 WT or mutant proteins were mixed with DSR2 protein at the molar ratio 1:3 in 200 µl reaction buffer. The amylose resin (NEB, E8021L) was washed by the reaction buffer for 3 times for equilibration. 20 µl equilibrated resin was applied to each reaction mixture and rotated at 4 °C for 1 h to pellet the DSR2 and MBP-DSAD1 complexes. The target proteins were eluted from the resins with 15 mM D-Maltose dissolved in the reaction buffer after washed 4 times with the reaction buffer. The eluted proteins were boiled with 2X SDS−PAGE loading buffer and then separated by SDS-PAGE and visualized by Coomassie Brilliant Blue staining.

## Strep pulldown assay

The competitive binding of MBP-DSAD1 and Strep-tagged TTP to the DSR2 protein was analyzed in a reaction buffer containing 25 mM Tris, pH 7.5, 100 mM NaCl, and 1 mM DTT. Initially, 50 µg of DSR2 proteins were either pre-mixed with TTP at a 1:1 molar ratio or mixed with DSAD1 at the indicated molar ratio in a 200 µl reaction buffer. Following a 1-h incubation on ice, the third protein (DSAD1 or TTP) was added to the reaction mixtures, which were then incubated for an

additional 1 h on ice. To achieve equilibration, the Strep-Tactin resin (NUPTEC, NRPB42L) was washed three times with the reaction buffer. Subsequently, 15 µl of the equilibrated resin was added to each reaction mixture and rotated at 4 °C for 1 h to promote the pelleting of the protein complexes. After being washed four times with the reaction buffer, 30 µl of 1X SDS-PAGE loading buffer was directly added to the resin. The samples were then boiled for 10 min, separated by SDS-PAGE, and detected by Coomassie Brilliant Blue staining.

## Cryo-EM data collection and image processing

Purified DSR2 (H171A) was crosslinked with 1 mM bis (sulfosuccinimidyl) suberate (BS3) in the presence of NAD$^+$ on ice for 30 min, and then loaded onto Superose 6 Increase 10/300 GL equilibrated with 20 mM HEPES pH 7.5, 100 mM NaCl, 0.5 mM TCEP. Crosslinked DSR2 was concentrated to 10.6 mg/mL and supplied with 0.01% IGEPAL® CA-630. The sample was applied onto glow-discharged holey carbon grids (Quantifoil Cu 300 mesh, R1.2/1.3). The grids were blotted for 3.5 s at 100% humidity using a Vitrobot Mark IV System (Thermo Fisher Scientific) before plunging into liquid ethane.

Purified DSR2 (H171A)-TTP complex was crosslinked with 1 mM BS3 on ice for 30 min, and then loaded onto Superose 6 Increase 10/300 GL equilibrated with 20 mM HEPES pH 7.5, 100 mM NaCl, 0.5 mM TCEP. Crosslinked DSR2-TTP was concentrated to 10 mg/mL and supplied with 1 mM NAD$^+$ and 0.01% IGEPAL® CA-630. The sample was applied onto glow-discharged holey carbon grids (Quantifoil Cu 300 mesh, R1.2/1.3). The grids were blotted for 3.5 s at 100% humidity using a Vitrobot Mark IV System (Thermo Fisher Scientific) before plunging into liquid ethane.

Purified DSR2-DSAD1 complex at 0.5 mg/mL was supplied with 0.01% IGEPAL® CA-630 and then applied onto glow-discharged holey carbon grids with ultrathin carbon layer (Quantifoil Cu 300 mesh + 2 nm C, R1.2/1.3). The grids were blotted for 3.5 s and wait for 60 s at 100% humidity using a Vitrobot Mark IV System (Thermo Fisher Scientific) before plunging into liquid ethane. To obtain particles in different orientation, purified DSR2-DSAD1 complex was crosslinked with 1 mM BS3 on ice for 30 min, and then loaded on to Superose 6 Increase 10/300 GL equilibrated with 20 mM HEPES pH 7.5, 100 mM NaCl, 0.5 mM TCEP. Crosslinked DSR2-DSAD1 complex was concentrated to 7.7 mg/mL and supplied with 0.01% IGEPAL® CA-630. The sample was applied onto glow-discharged holey carbon grids (Quantifoil Cu 300 mesh, R1.2/1.3). The grids were blotted for 3.5 s at 100% humidity using a Vitrobot Mark IV System (Thermo Fisher Scientific) before plunging into liquid ethane. The girds were screened on a Glacios Cryo-TEM (Thermo Fisher Scientific).

4120 movie stacks for DSR2 (H171A) sample were collected on Titan Krios G4 Cryo-TEM (Thermo Fisher Scientific) operating at 300 kV equipped with a Falcon 4i Direct Electron Detector and Selectris X Imaging Filter at a nominal magnification of 130,000 × (corresponding to 0.92 Å/pixel) and an accumulated dose of 50 e⁻/Å². 3383 movie stacks for DSR2 (H171A)-TTP complex sample were collected on Titan Krios G4 Cryo-TEM (Thermo Fisher Scientific) operating at 300 kV equipped with a Falcon 4i Direct Electron Detector and Selectris X Imaging Filter at a nominal magnification of 105,000× (corresponding to 1.2 Å/pixel) and an accumulated dose of 50 e⁻/Å². Patch motion correction was performed in CryoSPARC[34]. 2,711 movie stacks for crosslinked DSR2-DSAD1 complex and 2,391 movie stacks for uncrosslinked DSR2-DSAD1 complex were collected on Titan Krios G3i Cryo-TEM (Thermo Fisher Scientific) operating at 300 kV equipped with a K3 Camera (Gatan) and BioQuantum K3 Imaging Filter (Gantan Model 1967) at a nominal magnification of 81,000 × (corresponding to 1.0773 Å/pixel) and an accumulated dose of 50 e⁻/Å². Motion correction was performed with the MotionCorr2 program[35]. Patch CTF estimation and following processing steps were performed in CryoSPARC. Micrographs with better than 10 Å resolution and relative ice

thickness ranging from 1 to 1.2 were selected for particle picking. Generally, particles were initially picked up with Topaz using a pretrained model[36]. Class averages representing projections in different orientations selected from the second round of 2D classification were used as templates for Topaz model training. Then the generated model was used to pick particles for whole datasets. Particles were sorted by 2D classification and particles with clear structural features were chosen for generating initial maps. After several cycles of *ab-intio* reconstruction and heterogeneous refinement, particles from the best classes of heterogeneous refinement with the density of entire or partial DSR2 tetramer were applied to non-uniform refinement followed by local refinement, generating high-resolution map for initial model building[37]. The particles in the classes of entire DSR2 tetramer in the dataset of DSR alone, DSR2-TTP complex or DSR2-DSAD1 complex were chosen for 3D variability analysis and following non-uniform and local refinement. The composite maps were generated in Chimera[38].

### Model building and refinement
AlphaFold2 predicted model of DSR2 was split into three parts and separately docked into high-resolution cryo-EM map of DSR2-DSAD1 complex in UCSF Chimera[38,39]. The initial model of DASD1 based on high-resolution map was generated by Map to Model in Phenix[40–42]. Initial models were iteratively manually built in Coot[43] and refined with real-space refinement in Phenix[44]. The refined model was docked to other cryo-EM maps, followed by manually building in Coot and real-space refinement in Phenix. Model validation was performed in Molprobity[41,45]. Data collection and processing, model refinement and statistics are shown in Supplementary Table 1. Representative cryo-EM images, 2D class averages and 3D maps of each complex are shown in Supplementary Figs. 2–5. Structural images were generated in UCSF Chimera[38] and PyMOL (Schrodinger).

### Reporting summary
Further information on research design is available in the Nature Portfolio Reporting Summary linked to this article.

## Data availability
The cryo-EM density maps generated in this study have been deposited in the Electron Microscopy Data Bank (EMDB) under accession code EMDB-38872 [https://www.emdataresource.org/EMD-38872] (DSR2 H171A (crosslinked) partial structure); EMDB-38824 [https://www.emdataresource.org/EMD-38824] (DSR2 H171A (crosslinked) apo structure); EMDB-39925 [https://www.emdataresource.org/EMD-39925] (DSR2-TTP-NAD$^+$ complex); EMDB-38889 [https://www.emdataresource.org/EMD-38889] (DSR2-DSAD1 (crosslinked) partial complex); EMDB-38902 [https://www.emdataresource.org/EMD-38902] (DSR2-DSAD1 complex with two DSAD1 on the same side); EMDB-38907 [https://www.emdataresource.org/EMD-38907] (DSR2-DSAD1 with two DSAD1 on the opposite side). The atomic coordinates have been deposited in the Protein Data Bank (PDB) with accession number 8Y34 [https://www.rcsb.org/structure/unreleased/8Y34] (DSR2 H171A (crosslinked) partial structure); 8Y13 [https://www.rcsb.org/structure/unreleased/8Y13] (DSR2 H171A (crosslinked) apo structure); 8ZC9 [https://www.rcsb.org/structure/unreleased/8ZC9] (DSR2-TTP-NAD$^+$ complex); 8Y3M [https://www.rcsb.org/structure/unreleased/8Y3M] (DSR2-DSAD1 (crosslinked) partial complex); 8Y3W [https://www.rcsb.org/structure/unreleased/8Y3W] (DSR2-DSAD1 complex with two DSAD1 on the same side); 8Y3Y [https://www.rcsb.org/structure/unreleased/8Y3Y] (DSR2-DSAD1complex with two DSAD1 on the opposite side). Source data are provided with this paper.

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

## Acknowledgements
We thank the staff at the Cryo-EM Facility of Westlake University for technical assistant. This work was supported in part by the National Natural Science Foundation of China (32370742 to F.L., 32271264 to Z.S., 32200129 to R.Y., 81901168 to L.Y.), Huxiang Young Talents Program (2022RC1163 to F.L.), Natural Science Foundation of Hunan Province (2022JJ20056 to F.L., 2023JJ40437 to R.Y., 2021JJ20075 to L.Y.), Natural Science Foundation of Changsha City (kq2202088 to F.L.) and the Westlake Education Foundation (to Z.S.).

## Author contributions
F.L., Z.S. and R.Y. conceived and supervised the project. R.W., Q.X., T.L., Z.W. and H.G. performed plasmids construction, protein purification, and biochemical analysis. Q.X., J.L. and Y.S. performed Cryo-EM grids preparation, data collection and structure data analysis. L.Y. participated in biochemical assay design and manuscript preparation. F.L., Z.S. and H.G. built the structural models and wrote the manuscript.

## Competing interests
The authors declare no competing interests.
