## [Peer Review File · Nature Communications]

Reviewers' Comments:

Reviewer #1:

Remarks to the Author:

Wang et al. report on the structural basis of inhibition of the Dsr2 defence protein by the phage-encoded DSAD1 protein. Structures of DSR2 and the complex with DSAD1 using cryoEM are presented, and conclusions tested by sdm.

DSR2 forms a tetrameric structure with considerable structural flexibility. This is an inactive form of the protein, which requires binding to the phage tail tube protein for activation. DSAD1 binds this complex in a 2:4 ratio. Key predicted interface residues are tested by sdm, confirming changes in affinity (see point 2 below).

The authors suggest DSAD1 binding induces inactivating structural changes in DSR2 and present a model based on their data, whereby DSAD1 displaces the phage tail-tube protein to inactivate DSR2 (Figure 5d) – see point 3 below.

Overall, there is some interesting structural data here, but the lack of supporting biochemical data, particularly activity data, reduces confidence in the conclusions drawn.

Major points.

1. The bulk of the experimental data in this manuscript consists of cryoEM data. This should be carefully checked by an expert reviewer.
2. The MBP-pull down experiment used to quantify the affinity between DSR2 and DSAD1 wt and variants does not yield data that allow a firm conclusion about the importance of any particular residue for binding. It would be far better to assay the activity of DSR2 in the presence of the various mutants. The authors mention the use of a well-established NADase assay for DSR2 but note "data not shown", which is not really acceptable. The authors seem to have everything they need to do these assays, even if the tail-tube protein is not suitable for structural analyses.
3. The authors have a structure for inactive DSAD1 and another for DSAD1 bound to DSR2. They postulate that DSAD1 displaces the tail tube protein from DSR2 to inactivate the protein. This poses a number of problems. Firstly, in the absence of an activated structure, it is impossible to say whether DSAD1 causes inactivating structural rearrangement in the enzymatic domain of DSR2 along the lines shown in figure 5. Secondly, it is not ruled out, and seems much more likely, that DSAD1 binds DSR2 and prevents binding of the tail tube protein, rather than displacing it as shown in figure 5d. After all, DSAD1 is likely expressed very early in infection and tail tube protein expressed later. To test this, activity studies with differential order of addition of components could be carried out to see whether tail tube protein is displaced by DSAD1 or just prevented from binding to DSR2.

Reviewer #2:

Remarks to the Author:

In this manuscript, Wang et al reports the inhibitory mechanism of the DSR2 NADase activity by DSAD1, using the cryo-EM method. However, I still have some comments:

1. The authors stated that the DSR2 H171A lost its ability to bind NAD, based on the fact that no NAD density was observed in the sample of DSR2 H171A with extra NAD. This conclusion is too arbitrary, which needs to be proved by measuring their affinity.
2. Then sentence "The HTH domain folds into two HTH motifs and a β hairpin between.' is incomplete.
3. Structural comparison of DSR2 and ThsA, Y134 and D135 are supposed to be important for NAD

coordination, this conclusion needs to be proved by the biochemical experiments.

4. In the provided PDB (6LHX), the structure is the apo form TshA, it is hard to identify the residues for NAD binding. The structure of NAD bound Sir2 in the Sir2-APAZ/pAgo was reported (PMID: 38200015), maybe the comparison of Sir2 in DSR2 and pAgo would be helpful.

5. In figure 2f and extended figure 1a, the results of gel filtration profile of DSR2 are inconsistent. The peak is 9 ml and 50 ml, please explain. Besides, it is better to use analytical ultracentrifugation to analyze the effect of the mutation on its oligomerization. Why R86E and Y260E (they are both in the interface 4) are eluted differently?

6. The author states that the oligomeric DSR2 lacks NAD hydrolysis activity, so why they used DSR2 H171A mutant to collect the cryo-EM data? If the DSR2 alone is inactivated, how does DSAD1 inhibit its NADase activity? Maybe DSAD1 and the phage tail protein (the activator) competitively bind DSR2? So, the author needs to test the binding affinity of DSR2 and DSAD1, and phage tail.

Reviewer #3:

Remarks to the Author:

Bacteria and archaea use an expanded set of antiviral defenses to protect against infection by phage. One such defense system, DSR2, uses an NAD⁺ depletion strategy through the activity of a SIR2 domain. In this manuscript, Wang and Xu et al. use cryo-EM to determine the 3D structure of a tetrameric DSR2 complex providing the first structural information regarding this defense system. The authors also reveal the structure of DSR2 in complex with the virally encoded anti-defense protein DSAD1 which was previously shown to inhibit DSR2 enzymatic activity. These structures suggest a mechanism for anti-defense by DSAD1 which likely locks DSR2 in a conformation which is not conducive to NAD⁺ binding and catalysis.

The results of this manuscript are clearly presented. The structural information provided in this manuscript adds to our understanding of NAD⁺ depleting prokaryotic defense systems which operate through oligomerization. The manuscript, while well organized, lacks data which would shore up some of the conclusions and make it overall a more comprehensive story. Importantly, NADase assays are absent in the current study but would be essential to explain how the oligomers observed in cryo-EM maps may be operating.

Suggested revisions:

- Missing word in introduction- "While in bacteria, Sir2 domain proteins recently identified to act as NADases to hydrolyze the bacterial cellular NAD⁺ to against phage infection." – Missing a word between to and against. Perhaps "protect"?
- The fact that NAD⁺ was not observed in the binding pocket of DSR2 may be due to the mutation but it is more than likely that NAD⁺ cannot bind until DSR2 is conformationally triggered by phage tail tube protein. This alternative interpretation comes in some form much later in the discussion but may be worth stating much earlier.
- Typo TshA -> TshA "To elucidate the mechanism of NAD⁺ binding, we conducted a detailed analysis of the overlaid structures of DSR2 and TshA."
- Can the authors more accurately/quantitatively describe the oligomeric state of the tetramer interface mutants? "The gel filtration profiles of the DSR2 mutant proteins exhibited slower migration compared to the wild-type protein, indicating disruption of DSR2 tetramerization (Fig. 2f)." If molecular weight standards were run this could indicate if the shifted peaks are representative of monomer or dimeric complexes.
- While the subject of the manuscript is the inactivation mechanism of DSAD1 on DSR2 as determined by cryo-EM, there is no NADase data presented showing any inactivation. It might be beneficial to understanding the mechanism if some NADase measurements could be made. At minimum, the impact of tetramer and dimer interface mutants on NADase activity should be explored to highlight the importance of these complexes on activity but also the impact of DSAD1 mutants on NADase inactivation could be explored to further emphasize the importance of these

interactions.

- Unless the authors can show it directly using mutagenesis, binding affinity measurements, and/or cryo-EM, the references to competition of binding to DSR2 by DSAD1 and tail tube should be removed. The model as it stands is incomplete without evidence of direct competition. The analysis should reflect this uncertainty- DSAD1 may bind to a completely separate site relative to the binding site for tail tube and therefore binding would not be considered competitive. For example, there is no evidence to suggest that DSAD1 dislodges the tail tube as seemingly presented in Fig. 5d.

Point-to-point response to reviewers' comments

Manuscript ID: NCOMMS-24-06231-T

Previous Title: The structural basis of DSAD1-DSR2 mediated phage immune evasion

Current Title: The structural basis of the activation and inhibition of DSR2 NADase by phage proteins

We express our gratitude to the reviewers for their valuable and constructive comments. We have addressed their comments raised during the previous round of review, resulting in significant improvements to the manuscript. Please find our point-by-point responses to the reviewers below. The reviewer's comments are presented in black text, while our responses are highlighted in blue.

REVIEWER COMMENTS

Reviewer #1 (Remarks to the Author):

Wang et al. report on the structural basis of inhibition of the Dsr2 defence protein by the phage-encoded DSAD1 protein. Structures of DSR2 and the complex with DSAD1 using cryoEM are presented, and conclusions tested by sdm.

DSR2 forms a tetrameric structure with considerable structural flexibility. This is an inactive form of the protein, which requires binding to the phage tail tube protein for activation. DSAD1 binds this complex in a 2:4 ratio. Key predicted interface residues are tested by sdm, confirming changes in affinity (see point 2 below).

The authors suggest DSAD1 binding induces inactivating structural changes in DSR2 and present a model based on their data, whereby DSAD1 displaces the phage tail-tube protein to inactivate DSR2 (Figure 5d) – see point 3 below.

Overall, there is some interesting structural data here, but the lack of supporting biochemical data, particularly activity data, reduces confidence in the conclusions drawn.

Response: We thank the reviewer for the excellent summary of our work and for the positive comments.

Major points.

1. The bulk of the experimental data in this manuscript consists of cryoEM data. This should be carefully checked by an expert reviewer.

Response: We carefully processed and presented our cryoEM results and the data was evaluated by other reviewers.

2. The MBP-pull down experiment used to quantify the affinity between DSR2 and DSAD1 wt and variants does not yield data that allow a firm conclusion about the importance of any particular residue for binding. It would be far better to assay the activity of DSR2 in the presence of the various mutants. The authors mention the use of a well-established NADase assay for DSR2 but note “data not shown”, which is not really acceptable. The authors seem to have everything they need to do these assays, even if the tail-tube protein is not suitable for structural analyses.

Response: This is a great suggestion. We have optimized our experiment system and carried out the *in vitro* NAD⁺ hydrolase assays to analyze the enzymatic activities of WT DSR2 and its mutants in the presence of the tail tube protein (TTP) (Updated Figs. 1b, j and Fig. 2f). TTP is essential for the NAD⁺ hydrolase activity of DSR, which is suppressed by the wildtype DSAD1 but less by its mutants, supporting our observation in the complex structure (Updated Figs. 5f and 6a, b).

3. The authors have a structure for inactive DSAD1 and another for DSAD1 bound to DSR2. They postulate that DSAD1 displaces the tail tube protein from DSR2 to inactivate the protein. This poses a number of problems. Firstly, in the absence of an activated structure, it is impossible to say whether DSAD1 causes inactivating structural rearrangement in the enzymatic domain of DSR2 along the lines shown in figure 5. Secondly, it is not ruled out, and seems much more likely, that DSAD1 binds DSR2 and prevents binding of the tail tube protein, rather than displacing it as shown in figure 5d. After all, DSAD1 is likely expressed very early in infection and tail tube protein expressed later. To test this, activity studies with differential order of addition of components could be carried out to see whether tail tube protein is displaced by DSAD1 or just prevented from binding to DSR2.

Response: We appreciate the reviewer's insightful suggestion. In the revised manuscript, we successfully determined the cryo-EM structure of the DSR2-TTP-NAD⁺ complex (Updated Fig. 3). This new structure provides insights into the interactions between DSR2 and TTP, revealing that TTP occupies the same pocket on DSR2 as DSAD1, indicating a competitive binding scenario between DSAD1 and TTP. To investigate the mechanism by which DSAD1 affects the binding of the TTP to DSR2, we performed the NADase assay by introducing DSAD1 and TTP to DSR2 in different orders. The activity studies demonstrated that DSAD1 effectively suppresses the enzymatic activity of DSR2 triggered by TTP, regardless of whether DSR2 encounters the TTP or DSAD1 first (Updated Figs. 6a-b). However, TTP can not obviously activate DSR2 in the presence of DSAD1. Additionally, the subsequent pull-down assay revealed that DSAD1 significantly inhibits the binding of TTP to DSR2, even when DSR2 initially encounters TTP (Updated Fig. 6c). These new results suggest that DSAD1 is a potent competitor of TTP for DSR2 binding, although they may not occur in cells.

Reviewer #2 (Remarks to the Author):

In this manuscript, Wang et al reports the inhibitory mechanism of the DSR2 NADase activity by DSAD1, using the cryo-EM method. However, I still have some comments:

Response: We thank the reviewer for the constructive comments.

1. The authors stated that the DSR2 H171A lost its ability to bind NAD, based on the fact that no NAD density was observed in the sample of DSR2 H171A with extra NAD. This conclusion is too arbitrary, which needs to be proved by measuring their affinity.

Response: We appreciate the reviewer's insightful comments. We conducted isothermal titration calorimetry (ITC) experiments to determine the binding affinity of NAD⁺ to both DSR2 WT and H171A mutant (refer to the figure below). Surprisingly, no significant binding was detected for either WT DSR2 or H171A mutant. Considering that our new structure of DSR2-tail tube protein (TTP)-NAD⁺ complex revealed the presence of four NAD⁺ densities, whereas the DSR2 H171A structure showed no additional NAD⁺ density, it appears that DSR2 may not exhibit strong binding

to the NAD⁺ substrate in the absence of an activator protein. Consequently, we have revised the sentence to convey the following: "This observation suggests that in the absence of an activator protein, DSR2 may lose its ability to bind NAD⁺."

2. Then sentence "The HTH domain folds into two HTH motifs and a β hairpin between.' is incomplete.

Response: We appreciate the reviewer for pointing out our language error. We have rephrased the sentence "The HTH domain folds into two HTH motifs and a β hairpin between" to "The HTH domain folds into two HTH motifs with a β hairpin located between them".

3. Structural comparison of DSR2 and ThsA, Y134 and D135 are supposed to be important for NAD coordination, this conclusion needs to be proved by the biochemical experiments.

Response: This is a good point. We performed NADase assays to analyze the enzymatic activity of DSR2 Y134A, D135A, and H171A mutants. The results demonstrated that mutations of these key residues, which are conserved in ThsA, obviously reduced the activity of DSR2 (Updated Fig. 1j).

4. In the provided PDB (6LHX), the structure is the apo form ThsA, it is hard to identified the residues for NAD binding. The structure of NAD bound Sir2 in the Sir2-APAZ/pAgo was reported (PMID: 38200015), maybe the comparison of Sir2 in DSR2 and pAgo would be helpful.

Response: This is a great suggestion. During the revision process, the cryo-EM structure of ThsA in complex with NAD⁺ (PDB ID: 8BTP) was also been solved. Consequently, we incorporated the structural comparison of the DSR2 Sir2 domain with the Sir2 domains in ThsA (PDB ID: 8BTP) and Sir2-APAZ/pAgo (PDB ID: 8UAF) complex into the updated version of the manuscript (Updated Figs. 1g-i).

5. In figure 2f and extended figure 1a, the results of gel filtration profile of DSR2 are inconsistent. The peak is 9 ml and 50 ml, please explain. Besides, it is better to use Analytical ultracentrifugation to analysis the effect of the mutation on its oligomerization. Why R86E and Y260E (they are both in the interface 4) are eluted differently?

Response: We sincerely appreciate the reviewer's excellent suggestions. The discrepancy in the gel filtration profiles of DSR2 presented in original Fig. 2f (Updated Supplementary Fig. 4a) and original Supplementary Fig. 1a (Updated Supplementary Fig. 1a) is due to the utilization of different size of columns. Specifically, the gel filtration profile shown in original Fig. 2f was obtained using the Superdex™ 200 Increase size exclusion column with 24 mL resin volume, while the gel filtration profile in original Supplementary Fig. 1a was generated using the UNIONDEX 200PG 16/60 size extrusion column with 120 mL resin volume. We have now included this column information in the figure legends for clarity.

In addition, we conducted analytical ultracentrifugation (AUC) experiments to investigate the oligomerization state of WT DSR2 and its mutant proteins. The results demonstrated that DSR2 WT exists as a tetramer in solution, while the DSR2 Y260E mutant forms a dimer, confirming the findings from our previous gel filtration data. However, the AUC profile of DSR2 R86E exhibited a broad peak, and the calculated molecular weight was close to that of a monomer (130 kD). Additionally, the enzymatic activity assay revealed that the R86E mutation dramatically reduced the activity of DSR2 (refer to the figure below). These findings indicate that the R86E mutation likely impacts the folding of DSR2. Considering that the DSR2 Y260E mutant, which aligns with

our expectations and exhibits similar activity to the DSR2 WT protein, we have decided to exclude the gel filtration analysis results for the DSR2 R86E mutant in the revised manuscript.

a, Gel filtration profiles of DSR2 WT and R86E, Y260E mutant proteins on the Superdex™ 200 Increase size exclusion column. **b**, Analytical ultracentrifugation analysis the molecular weight of WT DSR2 and its R86E, Y260E mutant proteins. **c**, NAD^+ hydrolyase activities of WT DSR2 and its mutants in the presence of TTP.

6. The author state that the oligomeric DSR2 lacks NAD hydrolysis activity, so why they used DSR2 H171A mutant to collect the cryo-EM data? If the DSR2 alone is inactivated, how the DSAD1 inhibits its NADase activity? Maybe DSAD1 and the phage tail protein (the activator) competitive bind DSR2? So, the author needs to test the binding affinity of DSR2 and DASD1, and phage tail.

Response: We thank the reviewer for these great suggestions. In the previous version of our manuscript, we were unable to conduct the NAD^+ hydrolysis assay, making it unclear whether the DSR2 tetramer alone is active *in vitro*. Therefore, we utilized the DSR2 H171A mutant to collect data. However, in the revised version, we not only successfully performed the NAD^+ hydrolysis assay but also determined the structure of the DSR2-TTP- NAD^+ complex. The NADase assay demonstrated that DSR2 alone is inactive but can be activated by TTP. Additionally, the enzymatic activity of DSR2 triggered by TTP can be suppressed by DSAD1, irrespective of whether DSR2 encounters TTP or DSAD1 first (Updated Figs. 6a-b). Our new structure revealed that TTP binds to the same pocket on DSR2 as DSAD1, suggesting competitive binding between DSAD1 and TTP (Updated Fig. 3). Although we attempted to measure the binding affinities of DSR2 to DSAD1 and TTP using isothermal titration calorimetry (ITC), both TTP and DSAD1 proteins exhibited a tendency to aggregate, leading to inconclusive results. As an alternative approach, we

conducted a pull-down assay to compare the binding affinity of TTP and DSAD1 to DSR2. The results clearly demonstrated that DSAD1 effectively inhibits the binding of TTP to DSR2, even when DSR2 encounters TTP first (Updated Fig. 6c). Overall, our structural and biochemical analyses provide compelling evidence that DSAD1 competes with TTP for binding to DSR2 and subsequently suppresses its enzymatic activity.

Reviewer #3 (Remarks to the Author):

Bacteria and archaea use an expanded set of antiviral defenses to protect against infection by phage. One such defense system, DSR2, uses an NAD⁺ depletion strategy through the activity of a SIR2 domain. In this manuscript, Wang and Xu et al. use cryo-EM to determine the 3D structure of a tetrameric DSR2 complex providing the first structural information regarding this defense system. The authors also reveal the structure of DSR2 in complex with the virally encoded anti-defense protein DSAD1 which was previously shown to inhibit DSR2 enzymatic activity. These structures suggest a mechanism for anti-defense by DSAD1 which likely locks DSR2 in a conformation which is not conducive to NAD⁺ binding and catalysis.

The results of this manuscript are clearly presented. The structural information provided in this manuscript adds to our understanding of NAD⁺ depleting prokaryotic defense systems which operate through oligomerization. The manuscript, while well organized, lacks data which would shore up some of the conclusions and make it overall a more comprehensive story. Importantly, NADase assays are absent in the current study but would be essential to explain how the oligomers observed in cryo-EM maps may be operating.

Response: we thank the reviewer for the cogent summary of our work and overall positive assessment of our work.

Suggested revisions:

1. Missing word in introduction- “While in bacteria, Sir2 domain proteins recently identified to act as NADases to hydrolyze the bacterial cellular NAD⁺ to against phage infection.” – Missing a word between to and against. Perhaps “protect”?

Response: We appreciate the reviewer for bringing this language error to our attention. We have now incorporated the word "protect" into the sentence.

2. The fact that NAD⁺ was not observed in the binding pocket of DSR2 may be due to the mutation but it is more than likely that NAD⁺ cannot bind until DSR2 is conformationally triggered by phage tail tube protein. This alternative interpretation comes in some form much later in the discussion but may be worth stating much earlier.

Response: This is a great suggestion. Based on our new biochemical and structural results, we have incorporated the sentence "This observation suggests that in the absence of an activator protein, DSR2 may lose its ability to bind NAD⁺" into the second paragraph of the Results section in the updated manuscript.

3. Typo TshA -> ThsA "To elucidate the mechanism of NAD⁺ binding, we conducted a detailed analysis of the overlaid structures of DSR2 and TshA."

Response: We thank the reviewer for pointing out this typo in the manuscript which has been corrected.

4. Can the authors more accurately/quantitatively describe the oligomeric state of the tetramer interface mutants? "The gel filtration profiles of the DSR2 mutant proteins exhibited slower migration compared to the wild-type protein, indicating disruption of DSR2 tetramerization (Fig. 2f)." If molecular weight standards were run this could indicate if the shifted peaks are representative of monomer or dimeric complexes.

Response: This is a good point. We have performed analytical ultracentrifugation (AUC) experiments to accurately measure the molecular weights of WT DSR2 and its mutants. The results demonstrated that WT DSR2 exists as a tetramer in solution, while the DSR2 Y260E mutant forms a dimer, which is highly consistent with the previous gel filtration data. However, the AUC profile of DSR2 R86E exhibited a broad peak, and the calculated molecular weight was close to that of a monomer (130 kD). Additionally, the enzymatic activity assay revealed that the R86E mutation

dramatically reduced the activity of DSR2 (refer to the figure below). These findings indicate that the R86E mutation is likely to disrupt the folding of DSR2. Considering that the DSR2 Y260E mutant, which aligns with our expectations and exhibits similar activity to WT DSR2, we have decided to exclude the gel filtration analysis results for the DSR2 R86E mutant in the revised manuscript.

a, Gel filtration profiles of DSR2 WT and R86E, Y260E mutant proteins on the Superdex™ 200 Increase size exclusion column. **b**, Analytical ultracentrifugation analysis the molecular weight of WT DSR2 and its R86E, Y260E mutant proteins. **c**, NAD⁺ hydrolyase activities of WT DSR2 or its mutants in the presence of TTP.

5. While the subject of the manuscript is the inactivation mechanism of DSAD1 on DSR2 as determined by cryo-EM, there is no NADase data presented showing any inactivation. It might be beneficial to understanding the mechanism if some NADase measurements could be made. At minimum, the impact of tetramer and dimer interface mutants on NADase activity should be explored to highlight the importance of these complexes on activity but also the impact of DSAD1 mutants on NADase inactivation could be explored to further emphasize the importance of these interactions.

Response: We appreciate the reviewer's insightful suggestion. In the revised manuscript, we carried out the NAD⁺ hydrolyase assay to assess the inhibitory capacity of WT DSAD1 or its mutants on the enzymatic activity of DSR2 in the presence of TTP. The activity assay revealed that DSAD1 mutants, which exhibit defects in binding to DSR2, abolished the inhibition of DSR2 NADase activity triggered by TTP (Updated Fig. 5f). Additionally, we analyzed the NADase activity of DSR2 mutants that disrupt the dimer interfaces (L1000A/M1001A and N202A) or the tetramer interface (Y260E) of DSR2 (Updated Fig. 2f). The results indicated that the dimerization

of DSR2 is crucial for its enzymatic activity, while the tetramer interface is dispensable for its activation.

6. Unless the authors can show it directly using mutagenesis, binding affinity measurements, and/or cryo-EM, the references to competition of binding to DSR2 by DSAD1 and tail tube should be removed. The model as it stands is incomplete without evidence of direct competition. The analysis should reflect this uncertainty- DSAD1 may bind to a completely separate site relative to the binding site for tail tube and therefore binding would not be considered competitive. For example, there is no evidence to suggest that DSAD1 dislodges the tail tube as seemingly presented in Fig. 5d.

Response: We thank the reviewer for these constructive comments. During the revision process, we made remarkable progress in this project by successfully determining the cryo-EM structure of the DSR2-TTP-NAD⁺ complex (Updated Fig. 3). This new structure unveils that TTP binds to the identical pocket on DSR2 as DSAD1, suggesting a competitive binding interaction between DSAD1 and TTP to DSR2. To confirm the competitive binding mode, we conducted the NADase assay with varying orders of DSAD1 and TTP addition to DSR2. The activity studies demonstrated that DSAD1 effectively suppresses the enzymatic activity of DSR2 triggered by TTP, irrespective of whether DSR2 encounters TTP or DSAD1 first (Updated Figs. 6a-b). However, TTP can not obviously activate DSR2 in the presence of DSAD1. In addition, the subsequent pull-down assay revealed that DSAD1 significantly inhibits the binding of TTP to DSR2, even when DSR2 is pre-incubated with TTP (Updated Fig. 6c). In all, our structural and biochemical findings support two models. Firstly, if the DSAD1 protein is expressed first upon bacterial infection by phages, DSAD1 binds to DSR2 and locks it in an inactive state, preventing abortive infection. On the other hand, if TTP is expressed before DSAD1 and forms the DSR2-TTP complex, DSAD1 displaces TTP from DSR2, leading to the inhibition of its enzymatic activity. As we do not have information regarding which protein is expressed first after infection, we have revised our model by excluding the possibility of DSAD1 displacing TTP from DSR2.

Reviewers' Comments:

Reviewer #1:

Remarks to the Author:

The authors have provided significant new experimental data which strengthen the paper and support the overall conclusions of the study.

My remaining comments relate mostly to the NADase assays, which are a welcome addition to the paper.

Figure 2f, 5f, 6a and 6b: The statistical analysis and sample number should be stated in the legend as well as the methods. For figure 2f, the concentration of each protein in the assay should also be stated.

The method used for the NAD hydrolase assay involved assembly of all components, incubation for 15 min and then transfer to the plate reader where fluorescence was measured. The reaction does not appear to be stopped. This means that the reaction is still proceeding during measurement, but it seems only a single time point is measured to generate the data shown in Figs 2, 5 and 6. It would be preferable to measure a rate of NAD degradation using this assay. Although I am not suggesting that the authors go back and do this for all their assays, could the authors comment on why they chose a single 15 min time point? For example, was this in the linear part of the reaction curve? Furthermore, the raw data are not provided for any of the NADase assays.

Some English language checking is required as there are still some typos in the text. For example "raction" and "palte" in lines 41-5-422.

Reviewer #2:

Remarks to the Author:

In the revised version of manuscript, Wang et al added the cryo-EM structure of DSR2-TTP in complex with NAD⁺. However, there are some vital drawbacks must be clarified before it accept for publication.

Major concerns:

1. line 280-281, the authors stated that the structural rearrangement induced the binding of DASD1 disrupted the binding ability of DSR2 to NAD⁺, however, in the recent publication in Nature Comm (PMID: 38555355), the results of Zhang et al suggested that DSR2-DASA1 can still bind NAD⁺ (PDB: 8WYF DSR2-DSAD1-NAD), which contrasts with this work. How does the author explain it?
2. In Fig.6d, it seems that the catalytic residue H171 did not occurred an obvious conformational change, so, how to explain the NADase activation by the TTP and the inhibition by DSAD1?

Minor concerns:

1. line 60, I could not understand the sentence "and DSR, among others"?
2. line 103, the result of the ITC should be added in to support their conclusion.
3. line 103-104, The authors stated Y134, D135, and H171 are the putative catalytic residues, in my opinion, in the Sir2 domain, a conserved Histidine and Asparagine are supposed to be the catalytic dyad.
4. line 184, Y114, D115 (not D114) and H152. Please note H152 and N112 are the catalytic residues.
5. Two articles on the structural basis for DSR2 NADase activated and inhibited were published in Nat Comm (38555355, 38538592), which should be cited.

Reviewer #3:

Remarks to the Author:

The authors provided excellent new data including NADase assays and a new cryo structure which clearly support their prior claims regarding competition for binding and the importance of different interfaces for proper folding and activity. I am satisfied with these new data and the analysis provided. I would recommend only that the R86E DSR2 activity assay, size exclusion, and analytical ultracentrifugation data be included as supplementary in the final published version since it was collected by the authors and shared in the rebuttal. There is always room for data like these.

Point-to-point response to reviewers' comments

Manuscript ID: NCOMMS-24-06231-A

Title: The structural basis of the activation and inhibition of DSR2 NADase by phage proteins

We sincerely thank the reviewers for their insightful and valuable comments. We have carefully addressed their comments raised during the second round of review, which has greatly improved our manuscript. Please find our point-by-point responses to the reviewers below. The reviewer's comments are presented in black text, while our responses are highlighted in blue.

REVIEWER COMMENTS

Reviewer #1 (Remarks to the Author):

The authors have provided significant new experimental data which strengthen the paper and support the overall conclusions of the study.

Response: We thank the reviewer for the constructive and positive comments.

My remaining comments relate mostly to the NADase assays, which are a welcome addition to the paper.

1. Figure 2f, 5f, 6a and 6b: The statistical analysis and sample number should be stated in the legend as well as the methods. For figure 2f, the concentration of each protein in the assay should also be stated.

Response: We sincerely appreciate the reviewer's constructive suggestions. We have modified the figure legends of Figures 2f, 5f, 6a, and 6b to state the statistical analysis and sample numbers. Additionally, the concentration of each protein in the assay shown in Figure 2f has been added to the figure legend.

2. The method used for the NAD hydrolase assay involved assembly of all components, incubation for 15 min and then transfer to the plate reader where fluorescence was measured. The reaction

does not appear to be stopped. This means that the reaction is still proceeding during measurement, but it seems only a single time point is measured to generate the data shown in Figs 2, 5 and 6. It would be preferable to measure a rate of NAD degradation using this assay. Although I am not suggesting that the authors go back and do this for all their assays, could the authors comment on why they chose a single 15 min time point? For example, was this in the linear part of the reaction curve? Furthermore, the raw data are not provided for any of the NADase assays.

Response: We appreciate the reviewer's insightful comments. We agree with the reviewer that the reaction time for the NADase assays should be within the linear part of the reaction curve. To address this, we have performed the NADase assay at different time points to help us choose the appropriate time point. The following results indicate that the 15-minute time point is still within the linear range. In addition, we have included all the raw data of the NADase assays as a source data file in the revised manuscript.

The NADase assay was employed to analyze the NAD⁺ hydrolase activity of DSR2 in the presence of TPP at different time point. In this assay, 1 μ M of DSR2 WT protein was preincubated with 8 μ M TPP for 30 minutes. The complex protein was added into reaction mixture containing 50 μ M ϵ -NAD⁺ at different time points, and then incubated at 37 °C for reaction. The fluorescence intensity was measured using the Bio Tek Synergy H1 Plate Reader. All assays were performed in triplicate (mean \pm SD, n = 3 independent replicates), and the standard deviations were calculated using GraphPad Prism.

3. Some English language checking is required as there are still some typos in the text. For example “raction” and “palte” in lines 41-5-422.

Response: We thank the reviewer for pointing out these typos in the manuscript which has been corrected in the revised manuscript.

Reviewer #2 (Remarks to the Author):

In the revised version of manuscript, Wang et al added the cryo-EM structure of DSR2-TTP in complex with NAD⁺. However, there are some vital drawbacks must be clarified before it accept for publication.

Response: We thank the reviewer for these insightful comments, which have greatly helped us improve our manuscript.

Major concerns:

1. line 280-281, the authors stated that the structural rearrangement induced the binding of DASD1 disrupted the binding ability of DSR2 to NAD⁺, however, in the recent publication in Nature Comm (PMID: 38555355), the results of Zhang et al suggested that DSR2-DASA1 can still bind NAD⁺ (PDB: 8WYF DSR2-DSAD1-NAD), which contrasts with this work. How does the author explain it?

Response: We sincerely appreciate the reviewer for pointing out this key question. We compared the DSR2-DSAD1 structure, as well as apo DSR2 structure, with that of DSR2-TTP-NAD⁺, we did not find obvious conformational change in DSR2 Sir2 domain and the NAD⁺ binding pocket, suggesting NAD⁺ can still bind to DSR2 in apo and DSAD1-bound state. The NAD⁺ binding to DSR2-DSAD1 complex is supported by the reported cryo-EM structure of the DSR2-DSAD1-NAD⁺ complex as the reviewer mentioned. Therefore, we deleted the statement “This structural rearrangement potentially disrupts the binding ability of DSR2 to NAD⁺” as it was based on our previous speculation that has been contradicted by the new structural evidence. Additionally, we have rewritten this part to clarify the activation mechanism of DSR2 (lines 273-299 in the revised manuscript). Briefly, the binding of the activator TTP to DSR2 induced conformational changes in the C-terminal cavities, and this signal will transmit to DSR2 Sir2 domain through the motion of helix-turn-helix (HTH) domain for activation. Although the DSR2-DSAD1 complex can still bind to NAD⁺, and the conformation of the C-terminal cavities of DSR2 in this complex has some

variations compared to the DSR2-apo structure, the HTH domain of DSR2 in the DSR2-DSAD1 complex and the DSR2-apo structure shows no obvious variation. This suggests that the inhibitor DSAD1 locks DSR2 in an inactive state, preventing its activation.

2. In Fig.6d, it seems that the catalytic residue H171 did not occur an obvious conformational change, so, how to explain the NADase activation by the TTP and the inhibition by DSAD1?

Response: In agreement with the reviewer's comment, we did not find obvious conformational change in DSR2 Sir2 domain upon DSAD1 or TTP binding. The possible reason is that the structure that we captured is still not in fully active state in DSR2-TTP-NAD⁺ structure as we used the catalytically inactive mutant, H171A. This is consistent with the observation that the substrate NAD⁺ does not position at the catalytic site. It is possible that the conformation would change upon NAD⁺ contacting the catalytic residues in DSR2 Sir2 domain including N133 and H171. Although no dramatic conformational changes occur in Sir2 domain upon DSAD1 or TTP binding, we observed a large motion in the HTH domain of DSR2 in TTP-bound state compared to apo and DSAD1-bound states. This motion only occurs on the TTP-binding side in the DSR2-TTP structure. TTP, but not DSAD1, directly contacts DSR2 HTH domain. As the HTH domain of DSR2 is linked to and also contacts the Sir2 domain, the motion occurring in the HTH domain would transmit to the Sir2 domain and influence its conformation, which could activate DSR2. Further studies on the structure of DSR2-TTP-NAD⁺ complex in fully active state will provide deeper insights into DSR2 activation mechanism. Although the conformation of the C-terminal cavities of DSR2 in the DSR2-DSAD1 complex has some variations compared to the DSR2-apo structure, the HTH domain of DSR2 in the DSR2-DSAD1 complex and the DSR2-apo structure shows no obvious variation. This suggests that the binding of DSAD1 maintains DSR2 in an inactive state. Given that DSAD1 could displace TTP from binding to DSR2, DSAD1 locks the DSR2 in the inactive state, preventing its activation by inhibiting TTP binding.

Minor concerns:

1. line 60, I could not understand the sentence "and DSR, among others"?

Response: We appreciate the reviewer for pointing out our language error. We have rephrased the sentence “and DSR, among others” to “and DSR systems”

2. line 103, the result of the ITC should be added in to support their conclusion.

Response: As we mentioned above, we did not find obvious conformational change in DSR2 Sir2 domain and the NAD⁺ binding pocket in apo DSR2 and DSR2-DSAD1 complex when compared to that in DSR2-TTP-NAD⁺ structure, suggesting NAD⁺ can still bind to DSR2 in apo and DSAD1-bound state. As we only incubated DSR2 sample with NAD⁺ before size exclusion but not immediately before grid preparation, NAD⁺ could dissociate from DSR2 during purification as its low binding affinity that can not be detected by ITC. We think NAD⁺ can bind DSR2 in all three states. The no obvious binding of NAD⁺ to DSR2 in ITC experiment could be caused by the low sensitivity of ITC and also little heat change upon NAD⁺ binding. Due to no strong evidence supporting DSR2 cannot bind NAD⁺, we removed the statement “Although we added NAD⁺ to the protein sample prior to cryo-EM grid preparation, no discernible NAD⁺ density was observed in the putative binding pocket. This observation suggests that in the absence of an activator protein, DSR2 may lost its ability to bind NAD⁺.” The initial ITC results are not included in the manuscript either.

3. line 103-104, The authors stated Y134, D135, and H171 are the putative catalytic residues, in my opinion, in the Sir2 domain, a conserved Histidine and Asparagine are supposed to be the catalytic dyad.

Response: We agree with the reviewer’s comments that the conserved residues N133 and H171 in Sir2 domain of DSR2 are the catalytic dyad. We have modified the sentence “The putative catalytic residues (Y134, D135, and H171) in the Sir2 domain of DSR2” to “The key residues constituting the catalytic pockets (N133, Y134, D135, and H171) in the Sir2 domain of DSR2 are remarkable conserved ” in line 101-102 of the revised manuscript. Additionally, we have noted that the conserved residues N112 and H152 in ThsA are the catalytic dyad in lines 186-187.

4. line 184, Y114, D115 (not D114) and H152. Please note H152 and N112 are the catalytic residues.

Response: We thank the reviewer for pointing out our careless mistakes. We have corrected the sentence “ADPR points into a pocket formed by the catalytic residues Y114, D114, and H152, which correspond to Y134, D135, and H171 in DSR2” in lines 184-185 (original manuscript) to the “ADPR points into the catalytic pocket formed by the residues N112, Y113, D114, and H152, which correspond to N133, Y134, D135, and H171 in DSR2” in lines of 185-186 of the revised manuscript. In addition, we have noted that the conserved residues N112 and H152 in ThsA are the catalytic dyad in lines 186-187.

5. Two articles on the structural basis for DSR2 NADase activated and inhibited were published in Nat Comm (38555355, 38538592), which should be cited.

Response: This is an excellent suggestion. We have added citations to the three recently published articles that provide insights into the structural basis for the activation and inhibition of DSR2 NADase activity, which were published in Nature Communications (38555355, 38538592, 38729958), in the revised manuscript.

Reviewer #3 (Remarks to the Author):

The authors provided excellent new data including NADase assays and a new cryo structure which clearly support their prior claims regarding competition for binding and the importance of different interfaces for proper folding and activity. I am satisfied with these new data and the analysis provided. I would recommend only that the R86E DSR2 activity assay, size exclusion, and analytical ultracentrifugation data be included as supplementary in the final published version since it was collected by the authors and shared in the rebuttal. There is always room for data like these.

Response: We thank the reviewer for the positive assessment of our work. As suggested by the reviewer, we have included the activity assay, size exclusion, and analytical ultracentrifugation data for the DSR2 R86E mutant in the revised manuscript.

Reviewers' Comments:

Reviewer #1:

Remarks to the Author:

The authors have addressed all my remaining questions. This manuscript is now good-to-go.

Reviewer #2:

Remarks to the Author:

In revision, the authors have updated the manuscript and addressed all my concerns, in my opinion, the manuscript is read for publication.

Reviewer #3:

Remarks to the Author:

The authors have complied with suggested changes and the resulting manuscript is much improved from previous iterations. Statistical analysis is now made clear with figure legend adjustments and new data are better contextualized. This reviewer is sufficiently satisfied with the quality of the draft.

Point-to-point response to reviewers' comments

Manuscript ID: NCOMMS-24-06231-B

Title: The structural basis of the activation and inhibition of DSR2 NADase by phage proteins

REVIEWER COMMENTS

Reviewer #1 (Remarks to the Author):

The authors have addressed all my remaining questions. This manuscript is now good-to-go.

Response: We thank the reviewer for reviewing our manuscripts and providing the constructive and positive comments.

Reviewer #2 (Remarks to the Author):

In revision, the authors have updated the manuscript and addressed all my concerns, in my opinion, the manuscript is ready for publication.

Response: We sincerely appreciate the reviewer's insightful and valuable comments, which have greatly helped us improve our manuscript.

Reviewer #3 (Remarks to the Author):

The authors have complied with suggested changes and the resulting manuscript is much improved from previous iterations. Statistical analysis is now made clear with figure legend adjustments and new data are better contextualized. This reviewer is sufficiently satisfied with the quality of the draft.

Response: We thank the reviewer for the positive comments and valuable suggestions.